

**Abundance of volatile organic compounds and their role in ozone pollution management:**
**Evidence from multi-platform observations and model representations during the 2021–**
**2022 field campaign in Hong Kong**
Xueying Liu[1], Yeqi Huang[1], Yao Chen[1], Xin Feng[1], Yang Xu[1], Yi Chen[2], Dasa Gu[1], Hao Sun[1],
Zhi Ning[1], Jianzhen Yu[1,2], Wing Sze Chow[2], Changqing Lin[3], Yan Xiang[4], Tianshu Zhang[3,5],
Claire Granier[6,7], Guy Brasseur[8,9], Zhe Wang[1*], Jimmy C. H. Fung[1,10*]
[1]Division of Environment and Sustainability, The Hong Kong University of Science and
Technology, Kowloon, Hong Kong SAR, China
[2]Department of Chemistry, The Hong Kong University of Science and Technology, Kowloon,
Hong Kong SAR, China
[3]Key Laboratory of Environmental Optics and Technology, Anhui Institute of Optics and Fine
Mechanics, Hefei Institutes of Physical Science, Chinese Academy of Sciences, Hefei, China
[4]Information Materials and Intelligent Sensing Laboratory of Anhui Province, Institutes of
Physical Science and Information Technology, Anhui University, Hefei, China
[5]Institute of Environment, Hefei Comprehensive National Science Center, Hefei, China
[6]NOAA Chemical Sciences Laboratory/CIRES, University of Colorado, Boulder, CO, USA
[7]Laboratoire d'Aerologie, CNRS, University of Toulouse UPS, Toulouse, France
[8]Environmental Modeling Group, Max Planck Institute for Meteorology, Hamburg, Germany
[9]Atmospheric Chemistry Observation & Modeling Laboratory, National Center for Atmospheric
Research, Boulder, CO, USA
[10]Department of Mathematics, The Hong Kong University of Science and Technology, Kowloon,
Hong Kong SAR, China
*Correspondence to*: Jimmy C. H. Fung (majfung@ust.hk) and Zhe Wang (z.wang@ust.hk)





**Abstract.** Volatile organic compounds (VOCs) are a diverse group of species that contribute to
ozone formation. However, our understanding of VOC dynamics and their effect on ozone
pollution is limited by the lack of long-term, continuous, and speciated measurements, especially
of oxygenated compounds. To address this gap, this study for the first time integrates on-land,
shipborne, and spaceborne measurements from a field campaign in Hong Kong during 2021–
2022, analyzing 45–98 VOC species over land and water. Results show that oxygenated VOCs
(OVOCs) account for 73% (37 ppbv) of the total VOC concentration and 56% of the total ozone
formation potential (OFP), underscoring their indispensable role in VOC chemistry. Despite such
importance, OVOCs are underestimated by 45%–70% in the CMAQ model, while non-methane
hydrocarbons (NMHCs) face a lesser underestimation of 47%–48% (i.e., "model
underestimation"). Meanwhile, the model does not currently account for 17–56 species of the
total measured VOCs (i.e., "model omission"). According to this, we break down the observed
overwater VOC concentration of 51 ppbv into three components: 9 ppbv (18%) successfully
represented, 35 ppbv (69%) underestimated, and 7 ppbv (14%) omitted in the model. For OFP,
the breakdown shows 26% successful representation, 54% underestimation, and 20% omission.
Together, both "omission" and "underestimation" reveal the overall "VOC underrepresentation"
in the model, which partly results in greater ozone sensitivity to VOCs than observed by
spaceborne TROPOMI in polluted areas. The findings provide valuable insights into regional
pollution dynamics, and inform VOC-related model development and air quality management.



## 1. Introduction

Volatile organic compounds (VOCs) in the atmosphere represent a diverse group of gaseous organic trace substances. This extensive family includes non-methane hydrocarbon VOCs (NMHCs) such as alkanes, alkenes, alkynes, and aromatics, as well as oxygenated VOCs (OVOCs) like aldehydes, alcohols, ketones, ethers, and organic peroxides, classified based on their molecular structures. VOCs are emitted from various anthropogenic and biogenic sources (Li et al., 2014; Guenther et al., 2012) and can also be produced through secondary atmospheric processes. In the ambient air, NMHCs can be oxidized to OVOCs, a process that often predominates the levels of OVOCs in many regions of China and globally (Chen et al., 2014; Liu et al., 2022). Once emitted or formed, VOCs are subject to photolysis by near-ultraviolet radiation, generating atmospheric radicals, or to oxidation by various oxidants including hydroxyl radicals ($\cdot$OH), nitrate radicals ($NO_3\cdot$), and ozone ($O_3$), resulting in the formation of secondary peroxyl radicals ($RO_x$). These processes are essential for atmospheric radical recycling and play a significant role to ozone and secondary organic aerosol formation (Wang W. et al., 2022a; Wang W. et al., 2024; Whalley et al., 2021).

Surface ozone, produced through photochemical reactions involving VOCs and nitrogen oxides ($NO_x$), poses risks to human health and ecosystem productivity. Between 2013 and 2019, surface ozone levels increased in several metropolitan regions of China. Although levels seemed to slightly decline around 2019 in areas such as the Beijing-Tianjin-Hebei (BTH), Yangtze River Delta (YRD), and Sichuan Basin (SCB), ozone levels in the Pearl River Delta (PRD) have continued to rise in recent years (Feng et al., 2023; Wang Y. et al., 2023a; Wang W. et al., 2022b; Wang W. et al., 2024). Ozone formation in these metropolitan urban clusters is found to be particularly sensitive to VOCs (Wang Y. et al., 2023a; Wang T. et al., 2022). Therefore, understanding VOC abundance and its role in regulating ozone is crucial in urban areas nationwide, especially in the PRD where both ozone and VOC levels have shown an upward trend (Wang Y. et al., 2023a; Feng et al., 2023).

Current VOC representation in chemical transport models remains largely suboptimal for several reasons. First, producing VOC emission inventories is more challenging than for $NO_x$ due to the larger number of compounds involved and the uncertain source profiles (Li et al., 2014; Mo et al., 2016). Volatile chemical product (VCP) emissions, which have gained increasing attention in urban areas, have only recently been incorporated into emissions inventory development (Coggon et al., 2021; Seltzer et al., 2021; Wang S. et al., 2024). To date, VOC emission inventories have performed poorly against observations on a global scale (von Schneidemesser et al., 2023; Rowlinson et al., 2023; Beaudry et al., 2025). Meanwhile, the complex chemistry of VOCs poses significant challenges for model representation, and efforts are underway to improve their parameterization through new chemical mechanisms and coefficients (Travis et al.,



2024; Bates et al., 2024; Bates et al., 2021; Zhu et al., 2024). Mostly importantly, unlike criteria
pollutants (i.e., $PM_{10}$, $PM_{2.5}$, ozone, $NO_2$, $SO_2$, and CO) that are routinely monitored, only a
limited number of VOC species are measured at a small number of sites or for episodic short
durations both in China and globally (Zhang et al., 2025; She et al., 2024; Ge et al., 2024; von
Schneidemesser et al., 2023). The lack of consistent, continuous measurements for many VOC
species, particularly OVOCs, hinders comprehensive VOC-related model evaluation and
improvement. This limitation impacts the assessment of ozone formation sensitivity to VOCs and
the formulation of emission reduction strategies for ozone pollution management. Therefore,
there is an urgent need to understand the extent to which VOCs are represented in models, and
how this representation affects the simulation of the ozone-precursor relationship.
Knowledge gaps remain in previous studies assessing the ability of chemical transport models to
reproduce the ambient abundance of VOCs. First, the scarcity of OVOC measurements has led
earlier research to focus primarily on NMHCs, with only a limited number of OVOCs
investigated (von Schneidemesser et al., 2023; Rowlinson et al., 2023; Ge et al., 2024). This
results in a gap in understanding the model representation of OVOCs, which is becoming
increasingly important as future emissions shift toward greater OVOC levels due to rising
solvent use and consumer products (Coggon et al., 2021; Karl et al., 2018). Secondly, many
studies report direct comparison between equivalent VOC species between observations and
models (Zhang et al., 2025; She et al., 2024; Ge et al., 2024; Zhu et al., 2024). However, models
often simulate only a subset of the total measured VOCs. The missing VOC reactivity in models
versus measurements can be partly attributed to these omitted species (Wang et al., 2024). Yet,
the extent of this omission remains largely unreported in previous studies, highlighting a gap in
quantifying the overall model representation of ambient VOC abundance and their reactivity.
To address these gaps, this study utilized a joint field campaign conducted by the Hong Kong
Environmental Protection Department (HKEPD) and the Hong Kong University of Science and
Technology (HKUST), which included a comprehensive suite of many first-time measurements.
It features the first shipborne mobile air quality observations over Hong Kong waters (Sun et al.,
2024; Xu et al., 2023), alongside year-round continuous ground-based measurements during
2021–2022 (Mai et al., 2024; Lin et al., 2021). Additionally, a wide range of VOC species was
measured, with 98 species (35 oxygenated) identified over water and 45 species (16 oxygenated)
on land. This study integrated field observations with spaceborne TROPOMI data to: (1) analyze
the temporal and spatial characteristics of ozone and its precursors, with a particular focus on
speciated VOCs, (2) quantify VOC representation in the CMAQ chemical transport model, and
(3) assess the impact of such VOC representation on the simulated ozone formation sensitivity.
This study represents the first attempt to integrate multi-source field observations, spaceborne
data, and chemical transport modeling to examine VOC and ozone pollution in South China. The



findings provide valuable insights into the regional VOC and ozone pollution dynamics, while also highlighting deficiencies in model VOC representation that are crucial for advancing VOC-related model development and air quality management.

## 2. Method

### 2.1 Surface measurements

#### 2.1.1 Land-based monitoring

The Hong Kong Environmental Protection Department (HKEPD) operates a routine monitoring network of 18 land-based Air Quality Monitoring Stations (AQMS), which include 14 general stations, 1 background station and 3 roadside stations, broadly distributed over the Hong Kong region (Fig. S1). Additionally, the Cape D'Aguilar Supersite (CDSS), a coastal background site located at Hok Tsui in the southeastern of Hong Kong Island, equipped with surface VOC measurements and ozone lidar, is also included in the AQMS network to represent land-based monitoring in this study.

While ozone and $NO_x$ were monitored across AQMS sites, VOC measurements were only available at three sites (i.e., general site Tung Chung, background site Hok Tsui, and roadside site Mong Kok; Fig. S1). The study used measurements from 2021 to 2022, using the following respective measurement instruments. First, we use hourly NO, $NO_2$, and ozone from the AQMS measured by various gas analyzers (Feng et al., 2023). For stations with online VOC measurements, 29 NMHC species were measured using an online gas chromatography (GC) system (Syntech Spectras GC955 series 611/811), including 11 alkanes, 9 alkenes, 1 alkyne, and 8 aromatics (see Table 1; Mai et al., 2024). In addition, daily OVOC samples were collected using 2,4-dinitrophenylhydrazine (2,4-DNPH) cartridges, and then analyzed for sixteen OVOCs species (see Table 1) using a high-performance liquid chromatography with ultra-violet spectroscopy (HPLC-UV).

In addition, a concurrent gridded sampling was conducted at 40 sites in the PRD region on September 4–5, 2022, to provide a snapshot of ground-based HCHO levels during autumn. This effort is part of a series of gridded sampling initiatives (Mo et al., 2023). Air samples were collected into 2,4-DNPH cartridges and subsequently analyzed with a HPLC system coupled with a mass spectrometry detector (Agilent G6400). Two samples were collected at each site: one in the morning (6:00–10:00) and the other in the afternoon (12:00–16:00). For the comparison with spaceborne TROPOMI in Sect. 5, we selected only the afternoon samples to align as closely as possible with the satellite's overpass time at 13:30 local time. For each HCHO sampling site, we matched a corresponding $NO_2$ value from the nearest station of the China





National Environmental Monitoring Center (CNEMC) to calculate the surface HCHO-to-NO$_2$
ratio for comparison with spaceborne ratio.
**2.1.2 Shipborne measurements**
Shipborne measurement cruises were conducted in Hong Kong coastal waters in different
seasons during 2021–2022, including spring (February 22–23 and April 23 in 2021), summer
(July 23 and July 27 in 2021) and fall (September 17 in 2021, and September 4–5 and November
13–14 in 2022). During these cruises, a highly integrated portable air station, i.e., MAS-AF300
(Sapiens), was developed to measure trace gases (including NO, NO$_2$, and ozone) at 1-minute
resolution (Che et al., 2020; Sun et al., 2016). To detect NMHCs, whole air canister samples
were collected using 2 L electropolished stainless-steel canisters and analyzed with a GC-
MSD/ECD/FID system, which detected 63 NMHCs (see Table 1; Sun et al., 2024). These
NMHCs are sampled hourly at a single point (i.e., point sampling), providing instantaneous
concentration measurements. Additionally, OVOC samples were collected using Sep-Pak silica
cartridge coated with acidified 2,4-DNPH (Waters, USA) and then analyzed by an ultra-high-
performance liquid chromatography (UHPLC) (Agilent 1290 Infinity II, USA) coupled with a
triple quadrupole ESI mass spectrometer (AB SCIEX QTRAP 4500 MS system, USA). This
method detected 35 OVOCs above the detection limit (see Table 1; Xu et al., 2023). Those VOC
samples were collected over a one-hour period (i.e., integrative sampling) and reported as hourly
averages. For model comparisons lacking sub-hour variability, we report the differences between
observed and modeled NMHCs and OVOCs using simulated hourly averaged concentrations.
**2.1.3 Ozone formation potential**
The ozone formation potential (OFP) scale has been extensively used to quantify the relative
effects of individual VOCs on ozone formation. We calculated OFP using the following formula,
$OFP_i = [VOC]_i \times MIR_i$
where $[VOC]_i$ is the concentration of ith VOCs (μg m$^{-3}$). $MIR_i$ is the maximum incremental
reactivity of i-th VOCs (g ozone g$^{-1}$ VOC) that quantifies how much additional ozone can be
produced from incremental increases of that specific VOC in the atmosphere. Some VOCs (e.g.,
formaldehyde and acetaldehyde) are major contributors to ·OH reactivity and ozone formation,
while some (e.g., benzaldehyde) contribute minimally to ·OH removal and inhibit ozone
formation (Zhang et al., 2019). Their contributions to ozone are represented by their respective
MIR values, as obtained from Carter (2010).





Table 1. List of measured NMHCs and OVOCs on shipborne and land-based platforms. Species present in the CMAQ model are highlighted in bold. Subscripts indicate two lumped species in the model: * for XYLMN (xylene and other polyalkyl aromatics except naphthalene) and # for ALDX (aldehydes with 3 or more carbons). Other species in bold are explicitly represented in the model.

| Shipborne measurements (98 VOCs) | | | | | | Land-based measurements (45 VOCs) | | | |
| --- | --- | --- | --- | --- | --- | --- | --- | --- | --- |
| 63 NMHCs (CMAQ: 10 species = 9 explicit + 1 lumped) | | | | 35 OVOCs (CMAQ: 7 species = 6 explicit + 1 lumped) | | 29 NMHCs (CMAQ: 9 species = 8 explicit + 1 lumped) | | 16 OVOCs (CMAQ: 5 species = 4 explicit + 1 lumped) | |
| **Ethane** | $C_2H_6$ | Cyclopentene | $C_5H_8$ | **Formaldehyde** | $CH_2O$ | **Ethane** | $C_2H_6$ | **Formaldehyde** | $CH_2O$ |
| **Propane** | $C_3H_8$ | **Isoprene** | $C_5H_8$ | **Acetaldehyde** | $C_2H_4O$ | **Propane** | $C_3H_8$ | **Acetaldehyde** | $C_2H_4O$ |
| i-Butane | $C_4H_{10}$ | 2-Methyl-1-Butene | $C_5H_{10}$ | **Propionaldehyde#** | $C_3H_6O$ | i-Butane | $C_4H_{10}$ | **Propionaldehyde#** | $C_3H_6O$ |
| n-Butane | $C_4H_{10}$ | 3-Methyl-1-Butene | $C_5H_{10}$ | **n+iso-Butyraldehyde#** | $C_4H_8O$ | n-Butane | $C_4H_{10}$ | **n+iso-Butyraldehyde#** | $C_4H_8O$ |
| i-Pentane | $C_5H_{12}$ | 2-Methyl-2-Butene | $C_5H_{10}$ | **Crotonaldehyde#** | $C_4H_6O$ | i-Pentane | $C_5H_{12}$ | **Crotonaldehyde#** | $C_4H_6O$ |
| n-Pentane | $C_5H_{12}$ | 1-Pentene | $C_5H_{10}$ | **Valeraldehyde#** | $C_5H_{10}O$ | n-Pentane | $C_5H_{12}$ | **Valeraldehyde#** | $C_5H_{10}O$ |
| 2,2-Dimethylbutane | $C_6H_{14}$ | trans-2-Pentene | $C_5H_{10}$ | **Isovaleraldehyde + 2/3-Pentanone#** | $C_5H_{10}O$ | 2-Methylpentane | $C_6H_{14}$ | **Isovaleraldehyde + 2/3-Pentanone#** | $C_5H_{10}O$ |
| 2,3-Dimethylbutane | $C_6H_{14}$ | cis-2-Pentene | $C_5H_{10}$ | **Hexaldehyde#** | $C_6H_{12}O$ | n-Hexane | $C_6H_{14}$ | **Hexaldehyde#** | $C_6H_{12}O$ |
| 2-Methylpentane | $C_6H_{14}$ | 3 + 4-Methyl-1-pentene | $C_6H_{12}$ | **Heptaldehyde#** | $C_7H_{14}O$ | n-Heptane | $C_7H_{16}$ | **Benzaldehyde#** | $C_7H_6O$ |
| 3-Methylpentane | $C_6H_{14}$ | 2-Methyl-1-pentene + 1-Hexene | $C_6H_{12}$ | **Octylaldehyde#** | $C_8H_{16}O$ | 2,2,4-Trimethylpentane | $C_8H_{18}$ | **o-Tolualdehyde#** | $C_8H_8O$ |
| n-Hexane | $C_6H_{14}$ | **Benzene** | $C_6H_6$ | **Nonanaldehyde#** | $C_9H_{18}O$ | n-Octane | $C_8H_{18}$ | **m-Tolualdehyde#** | $C_8H_8O$ |
| 2,3-Dimethylpentane | $C_7H_{16}$ | **Toluene** | $C_7H_8$ | **Decylaldehyde#** | $C_{10}H_{20}O$ | **Ethene** | $C_2H_4$ | **p-Tolualdehyde#** | $C_8H_8O$ |
| 2,4-Dimethylpentane | $C_7H_{16}$ | Styrene | $C_8H_8$ | **Undecanal#** | $C_{11}H_{22}O$ | Propene | $C_3H_6$ | **2,5-Dimethylbenzaldehyde#** | $C_9H_{10}O$ |
| 2-Methylhexane | $C_7H_{16}$ | Ethylbenzene | $C_8H_{10}$ | **Dodecanal#** | $C_{12}H_{24}O$ | 1-Butene | $C_4H_8$ | **Acrolein** | $C_3H_4O$ |
| 3-Methylhexane | $C_7H_{16}$ | i-Propylbenzene | $C_9H_{12}$ | **Tridecanal#** | $C_{13}H_{26}O$ | trans-2-Butene | $C_4H_8$ | **Acetone** | $C_3H_6O$ |
| n-Heptane | $C_7H_{16}$ | n-Propylbenzene | $C_9H_{12}$ | **Salicylicaldehyde#** | $C_7H_6O_2$ | cis-2-Butene | $C_4H_8$ | 2-Butanone | $C_4H_8O$ |
| 2,2,4-Trimethylpentane | $C_8H_{18}$ | **m + p-Xylene*** | $C_8H_{10}$ | **Benzaldehyde#** | $C_7H_6O$ | **1,3-Butadiene** | $C_4H_6$ | | |
| 2,3,4-Trimethylpentane | $C_8H_{18}$ | **o-Xylene*** | $C_8H_{10}$ | **o-Tolualdehyde#** | $C_8H_8O$ | **Isoprene** | $C_5H_8$ | | |
| 2-Methylheptane | $C_8H_{18}$ | **2-Ethyltoluene*** | $C_9H_{12}$ | **m/p-Tolualdehyde#** | $C_8H_8O$ | 1-Pentene | $C_5H_{10}$ | | |
| 3-Methylheptane | $C_8H_{18}$ | **3-Ethyltoluene*** | $C_9H_{12}$ | **2,5-Dimethylbenzaldehyde#** | $C_9H_{10}O$ | trans-2-Pentene | $C_5H_{10}$ | | |
| n-Octane | $C_8H_{18}$ | **4-Ethyltoluene*** | $C_9H_{12}$ | **Methacrolein#** | $C_4H_6O$ | **Benzene** | $C_6H_6$ | | |
| n-Nonane | $C_9H_{20}$ | **1,2,3-Trimethylbenzene*** | $C_9H_{12}$ | 2-Furaldehyde | $C_5H_4O_2$ | **Toluene** | $C_7H_8$ | | |
| n-Decane | $C_{10}H_{22}$ | **1,2,4-Trimethylbenzene*** | $C_9H_{12}$ | **Acrolein** | $C_3H_4O$ | Ethylbenzene | $C_8H_{10}$ | | |
| Cyclopentane | $C_5H_{10}$ | **1,3,5-Trimethylbenzene*** | $C_9H_{12}$ | **Acetone** | $C_3H_6O$ | **m + p-Xylene*** | $C_8H_{10}$ | | |
| Methylcyclopentane | $C_6H_{12}$ | Alpha-Pinene | $C_{10}H_{16}$ | 2-Butanone | $C_4H_8O$ | **o-Xylene*** | $C_8H_{10}$ | | |
| Cyclohexane | $C_6H_{12}$ | Beta-Pinene | $C_{10}H_{16}$ | Cyclopentanone | $C_5H_8O$ | **1,2,3-Trimethylbenzene*** | $C_9H_{12}$ | | |
| Methylcyclohexane | $C_7H_{14}$ | **Ethyne** | $C_2H_2$ | Cyclohexanone | $C_6H_{10}O$ | **1,2,4-Trimethylbenzene*** | $C_9H_{12}$ | | |
| **Ethene** | $C_2H_4$ | Propyne | $C_3H_4$ | 2-Hexanone | $C_6H_{12}O$ | **1,3,5-Trimethylbenzene*** | $C_9H_{12}$ | | |
| Propene | $C_3H_6$ | | | 4-Methyl-2-pentanone | $C_6H_{12}O$ | **Ethyne** | $C_2H_2$ | | |
| 1,2-Propadiene | $C_3H_4$ | | | 2-Nonanone | $C_9H_{18}O$ | | | | |
| i-Butene | $C_4H_8$ | | | Acetophenone | $C_8H_8O$ | | | | |
| 1-Butene | $C_4H_8$ | | | Glyoxal | $C_2H_2O_2$ | | | | |
| trans-2-Butene | $C_4H_8$ | | | **Methylglyoxal** | $C_3H_4O_2$ | | | | |
| cis-2-Butene | $C_4H_8$ | | | 2,3-Butanedione | $C_4H_6O_2$ | | | | |
| **1,3-Butadiene** | $C_4H_6$ | | | Acetanisole | $C_9H_{10}O_2$ | | | | |



**2.2 Satellite measurements: S5P TROPOMI**
The TROPOspheric Monitoring Instrument (TROPOMI) is a nadir viewing spectrometer
onboard Sentinel-5P (S5P) satellite, which was launched on October 13, 2017. It operates in sun-
synchronous, low-Earth (825 km) orbits, with an Equator overpass at approximately 13:30 local
solar time (Veefkind et al., 2012). TROPOMI measures column amounts of several trace gases in
the ultraviolet-visible-near-infrared (UV-VIS-NIR; e.g., $NO_2$ and HCHO) and shortwave infrared
(SWIR; e.g., CO) spectral regions. TROPOMI has a horizontal swath of 2600 km that is divided
into 450 across-track rows. The spatial resolution of TROPOMI at nadir is 3.5 km × 7 km
(across-track × along-track) for $NO_2$ and HCHO, which was later refined to 3.5 km × 5.5 km in
August 2019 due to an adjustment to the along-track integration time. This study used
TROPOMI Product Algorithm Laboratory (PAL) tropospheric vertical column densities (VCDs)
for a consistent reprocessed data product over 2021–2022 using the same operational processor
(Eskes et al., 2021). For quality assurance, only observations with the overall quality flag
(qa_value) > 0.75 and retrieved cloud fraction (cloud_fraction) < 0.3 were used.
To compare model simulations with satellite retrievals, we followed the method established in
previous studies (Douros et al., 2023; Goldberg et al., 2022; Zhang et al., 2020; Sun et al., 2025).
First, CMAQ simulated profiles were sampled at the TROPOMI overpassing time and location
of each measurement. We then interpolated model profiles to the vertical levels of satellite
retrievals. For consistency between TROPOMI and CMAQ simulated vertical profiles, we also
applied scene-dependent averaging kernels that describe the instrument vertical sensitivity to
changes in a trace gas and replaced *a priori* information from the TM5-MP model used in the
standard retrieval. Finally, we integrated from the surface up to the tropopause to calculate
column values.
**2.3 The CMAQ model**
The Community Multiscale Air Quality Modeling System (CMAQ) is a three-dimensional
Eulerian atmospheric chemistry and transport model that simulates atmospheric composition
throughout the troposphere. Four domains were set up with different horizontal resolutions at 27
km, 9 km, 3 km, and 1 km (Fig. 1). The model employed a terrain-following coordinate system
with 38 vertical layers extending from the surface to the 50 hPa pressure level (approximately 23
km height). Chemical boundary and initial conditions for the outmost domain were generated
from the seasonal average hemispheric CMAQ outputs archived on its official website. Science
configuration options include the CB6r3 gas-phase chemistry (Luecken et al., 2019), the AERO7
aerosol module (Appel et al., 2021), and the M3Dry dry deposition scheme (Pleim et al., 1984).
The anthropogenic emissions used are the HKEPD emission inventory scaled to 2019 for HK
region and scaled to 2021 for the PRD region, the high-resolution INTegrated emission inventory



of Air pollutants for China (INTAC) scaled to 2019 for the rest of China, and the 2018 Emissions
Database for Global Atmospheric Research (EDGAR) for regions outside of China. Biogenic
emissions are from the Model of Emissions of Gases and Aerosols from Nature (MEGAN) v3.1.
The meteorological inputs to drive CMAQ were from the Weather Research and Forecasting
(WRF) model. Meteorological boundary and initial conditions were the National Centers for
Environmental Prediction (NCEP) Final Analysis data at 0.25º×0.25º. Land surface data (e.g.,
land use, vegetation type, terrain elevation, etc.) were obtained from the U.S. Geological Survey
(USGS) terrain databases, overwritten by the 2003 land use data over the PRD region from the
Planning Department of the Hong Kong Government. WRF physics schemes include
Asymmetric Convective Model v2 (ACM2) for the planetary boundary layer physics, the Unified
Noah Land Surface Model (Noah-LSM) for the land surface, single-moment 3-class scheme for
cloud microphysics, the rapid radiative transfer model (RRTM) for longwave and shortwave
radiation processes, and the Kain–Fritsch scheme for cumulus convections.

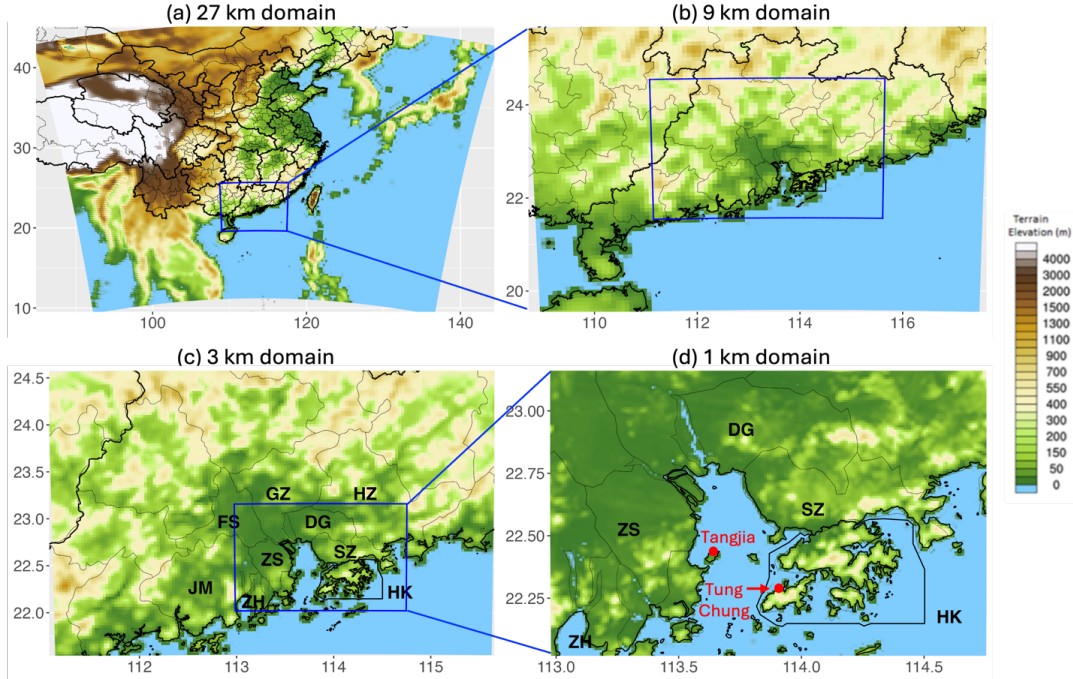

Figure 1. Model domains with terrain elevation. Abbreviated city names are inserted, include
Hong Kong (HK), Shenzhen (SZ), Dongguan (DG), Huizhou (HZ), Guangzhou (GZ), Foshan
(FS), Zhongshan (ZS), Jiangmen (JM), and Zhuhai (ZH).



**2.4 VOC comparisons between measurements and model**
As described in Sect. 2.1, land-based instruments measure a total of 45 VOC species (including
29 NMHCs and 16 OVOCs), while shipborne instruments measure 98 species (including 63
NMHCs and 35 OVOCs). These species represent the ambient measurable VOC abundance,
referred to as 'total observed' species. However, state-of-the-art chemical transport models do
not account for such a broad range of species. Instead, these models typically simulate key
species that significantly impact atmospheric composition individually (i.e., explicit species),
while grouping less critical species into a single representative species (i.e., lumped species) to
simplify the modeling process and improve computational efficiency. For instance, among all
measured species at land-based sites, only 13 NMHCs and 15 OVOCs (referred to as 'selective
observed' species) are represented in our CMAQ simulation by 9 NMHCs (8 explicit and 1
lumped species) and 5 OVOCs (4 explicit and 1 lumped species) (referred to as 'modeled'
species). Similarly, among the shipborne measured species, only 17 NMHCs and 25 OVOCs are
represented in CMAQ by 10 NMHCs (9 explicit and 1 lumped species) and 7 OVOCs (6 explicit
and 1 lumped species). See Table 1 for details. The differences between 'total observed' and
'selective observed' species reflect the abundance of species omitted by the model.
To evaluate the model's representation of VOCs, we conducted two types of observation–model
comparisons. First, we performed an equivalent comparison between 'modeled' and 'selective
observed' species to assess model bias in simulating the concentrations of VOC species that are
common to both the observations and the model. This helps identify discrepancies in the model's
ability to capture key species that significantly influence atmospheric chemistry, indicating either
underestimation or overestimation. Second, we compared 'modeled' to 'total observed' species
to evaluate the overall representativeness of the modeled VOCs in relation to the ambient
measurable VOC abundance. This comparison provides insight into the model's ability to reflect
the full range of VOCs present in the atmosphere.
**3. Ozone and precursors over land**
**3.1 Monthly variation**
Land-based measurements provide continuous surface monitoring across all seasons, which is
valuable for analyzing temporal variations in ozone and its precursors. Surface $NO_2$ levels
recorded at ground-based sites are higher in cold seasons than in warm seasons, with monthly
differences relative to the annual mean ranging from −18% in June–July-August (JJA) to +23%
in December–January–February (DJF) (Fig. 2a). This seasonal pattern of surface $NO_2$ correlates
well with the tropospheric vertical column densities (VCDs) of $NO_2$ measured by the spaceborne



TROPOMI (R = 0.9; Fig. S2), which show monthly differences relative to the annual mean
ranging from −20% in JJA to +60% in DJF over the Hong Kong domain. Comparing the
observed surface $NO_2$ with the CMAQ model (Fig. 2a), the model well captures the warm
seasons but underestimates $NO_2$ levels during the cold seasons, resulting in an overall
underestimation (R = 0.6; NMB = –30%).
Surface ozone in the subtropical coastal city of Hong Kong exhibits a bimodal pattern with a
strong peak in autumn and a minor peak in spring (Fig. 2b), which is also visible in boundary
layer ozone measurements recorded by the ozone lidar (Fig. S3). This bimodal pattern is
different from that of the North China Plain, where ozone typically peaks in the summer months
from May to July (Li et al., 2019; Wang W. et al., 2022b). In Hong Kong, the spring peak
consistently occurs in April for both the 2021–2022 average and the multi-decade average
(2000–2023). Springtime ozone exceedances are often attributed to the long-range transport of
air masses that are rich in ozone or its precursors from Southeast Asia (Wang et al., 2019; Lee et
al., 2019; Chan et al., 2000). The autumn peak occurs in September for the 2021–2022 average,
while the decadal average shows this peak in October. Late summer and autumn episodic events
are typically characterized by high temperatures and stagnant conditions associated with a
subtropical high or typhoon, which contribute to significant local ozone production (Lin et al.,
2021; Chen et al., 2024; Ouyang et al., 2022). Comparing observations with the model, the
model effectively captures the observed seasonal variation with some overestimation during cold
months (R = 0.8; NMB = 23% in Fig. 2b). This overestimation is lower when comparing
observed and modeled total oxidants ($O_x$) with improved metrics (R = 0.9; NMB = 2% in Fig.
2c).





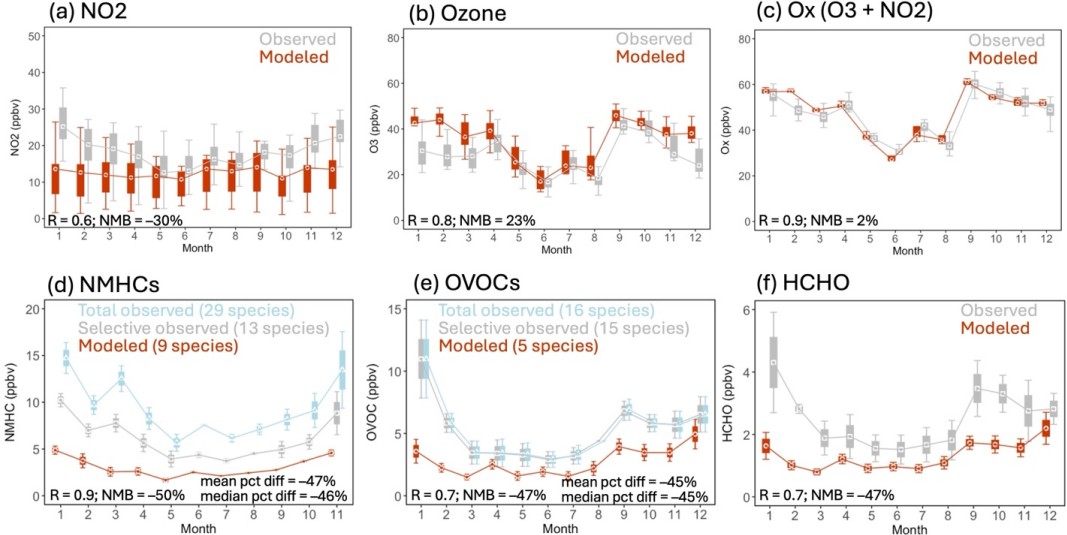

Figure 2. Monthly variations of different species recorded at land-based AQMS sites over 2021–2022. Error bars indicate variability across non-roadside sites. Correlation coefficient (R) and normalized mean bias (NMB) between model (in red) and observations (in grey) are inserted.

Land-based sites measure a total of 45 VOC species, including 29 NMHCs and 16 OVOCs. We observed that the concentration of NMHCs is higher in colder months compared to warmer months (Fig. 2d), primarily driven by anthropogenic species (e.g., ethene, ethane, propane), despite higher levels of the biogenic species (e.g., isoprene) in warmer months (Fig. S4). Meanwhile, OVOCs show peaks in January and September (Fig. 2e). HCHO, the most abundant atmospheric carbonyl primarily resulting from the secondary oxidation of various VOCs, follows a similar peak pattern (Fig. 2f). By comparing surface HCHO with spaceborne tropospheric columns (Fig. S2), we found that the September peak is evident in both surface and column observations, suggesting enhanced local secondary production from the oxidation of various VOCs, driven by Hong Kong's hot and sunny weather during this period. In contrast, the January peak is visible only at the surface but not in tropospheric columns. This January surface peak is associated with several days in 2021 at the Tung Chung site, where multiple VOCs spiked (Fig. S4), possibly due to a large anthropogenic emitter, causing an abrupt surge confined to the surface. Overall, the model captures the observed seasonal variations in NMHCs (Fig. 2d; R = 0.9; NMB = –50%), OVOCs (Fig. 2e; R = 0.7; NMB = –47%), and HCHO (Fig. 2f; R = 0.7; NMB = –47%).

A strong relationship exists between HCHO and ozone, as indicated by a strong correlation (R = 0.7) between the observed monthly variability of the two. In comparison, total VOCs including





both NMHCs and OVOCs, show only a modest correlation with ozone (R = 0.4), as do OVOCs
(R = 0.3) and NMHCs (R = 0.3). This strong HCHO–ozone relationship arises because HCHO
and ozone are coproduced through the photochemical oxidation of VOCs. The model effectively
captures the observed strong HCHO-ozone correlation (R = 0.6).
**3.2 HCHO as an indicator for ambient VOCs**
Many studies use the HCHO-to-$NO_2$ ratio (FNR) to indicate ozone formation sensitivity through
both surface in situ measurements (Tonnesen and Dennis, 2000; Liu et al., 2024) and spaceborne
remote sensing (Jin et al., 2017; Goldberg et al., 2022; Acdan et al., 2023). However, the
composition of VOCs varies significantly in different urban areas with specific emission profiles
and in biogenic-dominated natural environments. Therefore, one concern with the FNR method
is how representative HCHO is of ambient VOCs in different environments. To approach this
question and validate our FNR analysis in Sect. 5, we now assess this relationship in three
diverse environments (i.e., roadside, urban, and background) in Hong Kong.
First, we examine the relative contributions of HCHO to all measured VOCs at these locations.
Figure 3 shows that the contribution of HCHO to total VOC concentrations decreases from 31%
at roadside Mong Kok (Fig. 3a) to 21% at urban Tung Chung (Fig. 3d) and further to 14% at
background Hok Tsui (Fig. 3g). Similarly, the contribution of HCHO to total OFP decreases
from 35% at Mong Kok (Fig. 3b) to 24% at Tung Chung (Fig. 3e) and further to 20% at Hok
Tsui (Fig. 3h). Overall, HCHO is the largest component of all measured VOCs at both Mong
Kok and Tung Chung, indicating its key representativeness in polluted urban areas. At the
background site Hok Tsui, HCHO is the second-largest contributor, accounting for 14% of total
concentrations and 20% of OFP, which is slightly less than the contributions from the largest
contributor, ethane for concentration and isoprene for OFP. Finally, we examine the temporal
correlation between HCHO and total measured VOCs and find strong correlations at all sites:
R=0.81 at Mong Kok (Fig. 3c), R=0.85 at Tung Chung (Fig. 3f), and R=0.72 at Hok Tsui (Fig.
3i). VOC species that are omitted by the model account for 14%–17% of the total observed
concentrations and 16%–19% of their OFP across different environments.
The above analysis suggests that HCHO serves as a reliable indicator of ambient VOC
environments due to (1) its key role in representing VOC levels and OFP and (2) its strong
temporal correlations with the total measurable VOCs. However, its representativeness may
vary, being more pronounced in polluted areas and less so in natural settings. For surface in situ
measurements, which can effectively capture a diverse range of VOC species, a more
comprehensive understanding of local VOC environments can be achieved through the analysis
of multiple species. In the field of satellite remote sensing, HCHO is currently the most common





operation grade VOC retrievals from many instruments and often used as a single indictor to
represent overall VOC levels across different environments. Studies that use the FNR to identify
ozone formation regimes tend to filter out vast regions with low $NO_2$ values (Jin et al., 2020),
focusing instead on urban areas with high $NO_2$ levels, where HCHO is most representative of
ambient VOCs. As more satellite VOC products become available, such as glyoxal (CHOCHO)
(Lerot et al., 2021; Ha et al., 2024), isoprene ($C_5H_8$) (Wells et al., 2022), formic acid (HCOOH)
(Stavrakou et al., 2011), and ethane ($C_2H_6$) (Brewer et al., 2024), the validity of using a multi-
species representation of VOCs in various local conditions can be explored. This will aid in
ozone air quality studies across different environments with varying emission profiles and local
photochemistry.

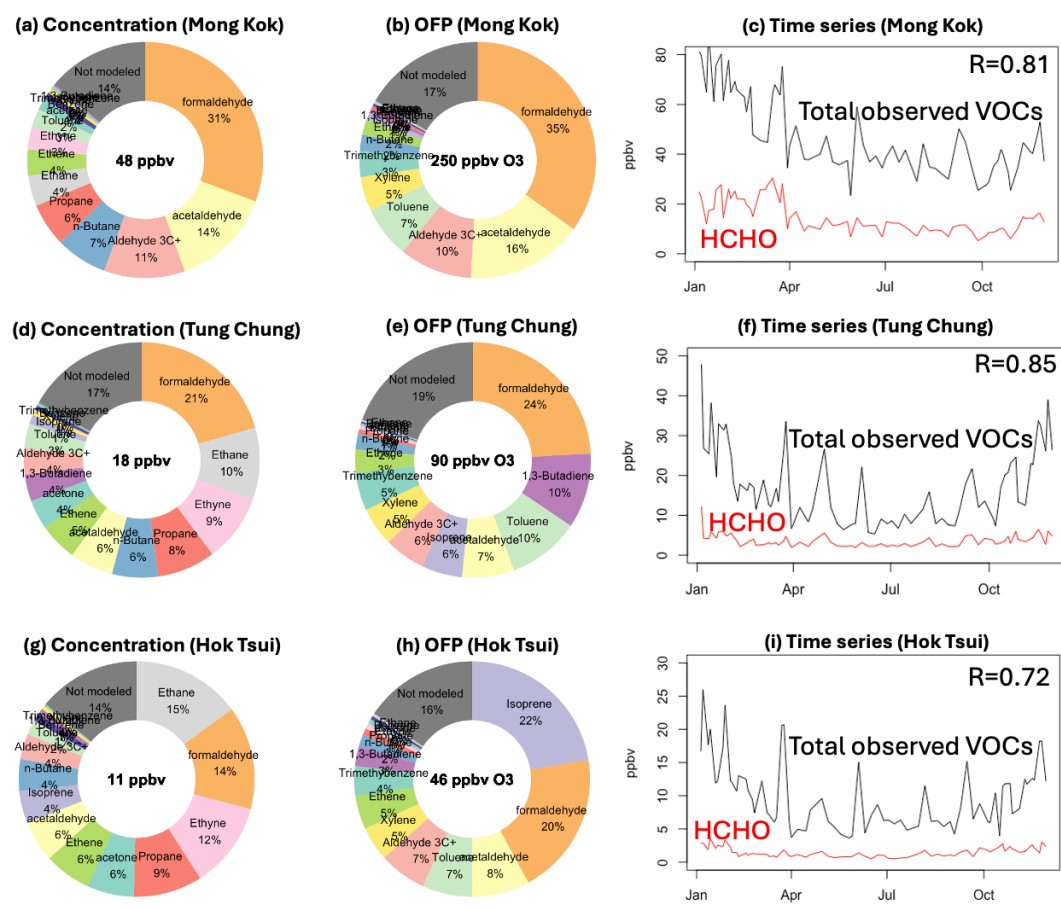

Figure 3. Contribution of individual VOCs to (a, d, g) the concentration and (b, e, h) the ozone
formation potential (OFP) of the total observed VOCs at three land-based sites in 2021. (c, f, i)
Time series of HCHO and total observed VOCs with their correlation coefficient inserted.





**4. Ozone and precursors over coastal waters**
**4.1 Spatial patterns**
Ozone and its precursors are measured in coastal waters for the first time during mobile ship
cruises in this region. High ozone levels were observed over water on July 23, July 27, and
September 17 in 2021, as well as on September 4–5 and November 13 in 2022 during the 10-day
ship cruises (Fig. 4a). On high-ozone days, the maximum hourly mean ozone concentrations
range from 91–196 ppbv, significantly higher than the 52–69 ppbv observed on clean days (Fig.
S6). These elevated levels were detected likely because the ships were specifically tasked with
measuring peak ozone episodes. Spatially, high ozone concentrations are generally found at the
eastern and western ends of the cruises, while much lower levels at around 20 ppbv are recorded
in the interchange between Kowloon Peninsula and Hong Kong Island due to the titration effect
of high $NO_x$ levels along intensive shipping routes (Wang Y. et al., 2023b). We examine
westward shipborne measurements into the Pearl River Estuary (PRE) and find that over-water
ozone concentrations are typically 12 ppbv higher than the land-based Tangjia site on the west
bank of the PRE and 32 ppbv higher than Tung Chung site on the east bank between 9 AM to 2
PM. This water-land gradient of ozone is more pronounced on high ozone days compared to
clean days (Fig. 5). Our findings, based on actual over-water measurements into the PRE for the
first time, confirm the PRE as an ozone pool simulated by Zeren et al. (2019).
For spatial pattern of $NO_2$, low concentrations are observed on eastbound routes (Fig. 4d), which
are generally upwind of Hong Kong and least affected by urban emissions. In contrast, high
concentrations were detected on westbound routes that passed through busy transportation and
shipping zones and reached the westernmost area of Hong Kong waters with influences by aged
plumes from the urban emissions in the PRD region. This east-west gradient in $NO_2$ is also
observed in mobile vehicle measurements reported by Zhu et al. (2018). The model reasonably
captures the overall observed east-west gradient, but tends to overestimate ozone and
underestimate $NO_2$ to the east of Hong Kong waters while exhibiting the opposite bias to the
west (Fig. 4c, 4f). To reflect the model's ability to simulate atmospheric oxidative capacity
without strong interreference from $NO_x$ titration effect, the oxidant level ($O_x = O_3 + NO_2$) is
often used as a tracer of photochemical processes in high $NO_x$ environment, such as along major
highways and shipping lanes during field campaigns (Li et al., 2023; Liu et al., 2024). By
comparing observed and modeled $O_x$, we find that the simulated $O_x$ has low bias and a
substantial correlation against observations (Fig. 4h), suggesting that the model effectively
captures regional oxidative capacity.





As introduced in Sect. 2.1.2, NMHCs were collected via point sampling at a single timestamp
within each hour, resulting in only a few isolated points being marked on the map for each daily
measurement period. OVOCs, on the other hand, were measured using integrative sampling over
the entire hour while the ship was in motion, resulting in no fixed latitude and longitude
coordinates assigned to them. As a result, there is no comparable spatial distribution of VOCs to
present here; instead, we directly compare their observation-model differences in Sect. 4.2.

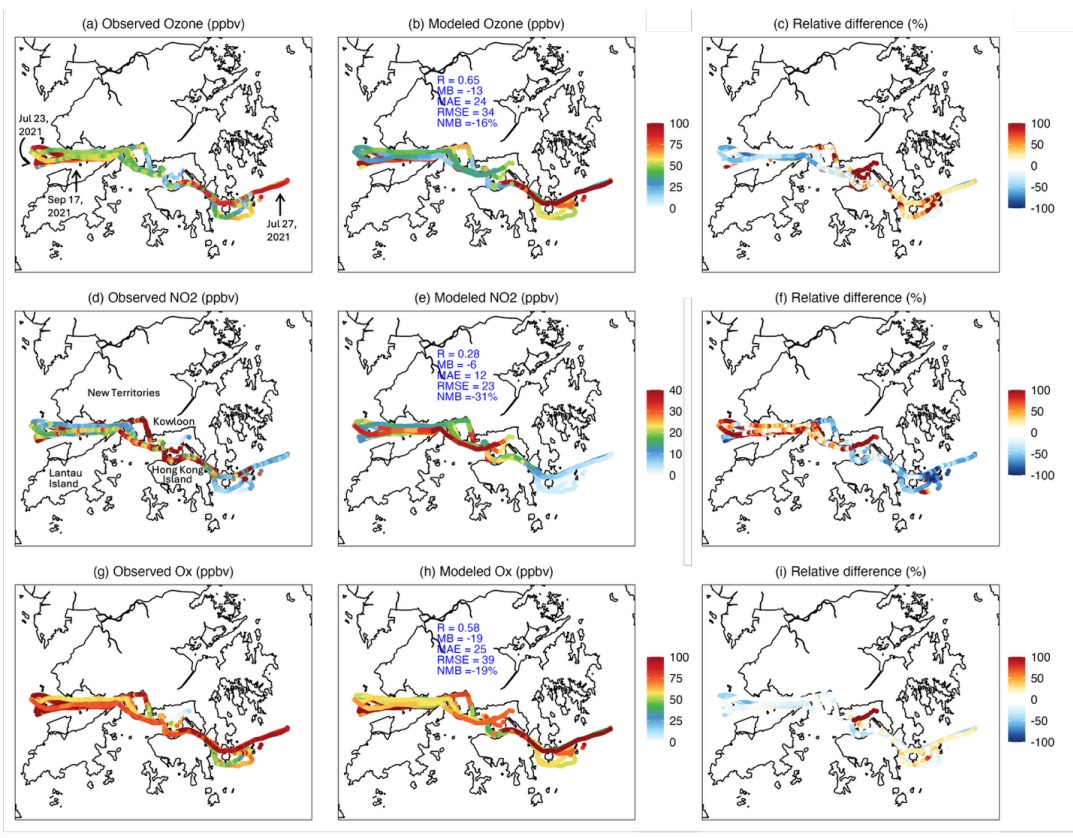

Figure 4. Mobile ship measurements of ozone (top), NO$_2$ (middle) and O$_x$ (bottom), along with
their modeled equivalents. Percentage differences are given as (model–observation)/observation.



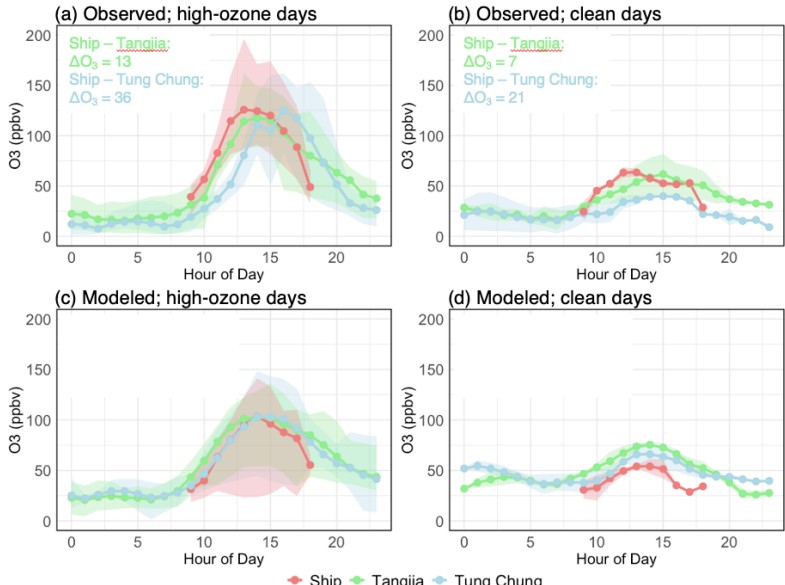

Figure 5. Ozone diurnal variations recorded during westward ship cruises and at two land-based
sites (Tangjia and Tung Chung; locations shown in Fig. 1). The mean water-land ozone gradient
(ΔO3) is calculated as shipborne ozone minus land-based measurements from 9 AM to 2 PM.

## 4.2 Underrepresented VOCs in the model

Two types of observation–model comparisons were conducted for VOCs, as described in Sect.
2.4. The first type compares equivalent VOC species common to both observations and the
model in Figure 6. We found that the model struggles to simulate medium to high concentrations
and thus a majority of the values cluster at low mixing ratios for both NMHCs (Fig. 6c) and
OVOCs (Fig. 6d). Notably, the model underestimates equivalent OVOCs by 76% and NMHCs
by 50%, significantly larger than that of ozone (–11%) and $NO_2$ (–14%) based on the medians of
their observation–model percentage differences (Fig. 6e). Such underestimation of VOCs has
been observed in various chemical transport models and mechanisms (She et al., 2024; Wang W.
et al., 2024; Ge et al., 2024; Chen et al., 2019; Rowlinson et al., 2024), though these studies
primarily focus on NMHCs and provide limited evidence for OVOCs.





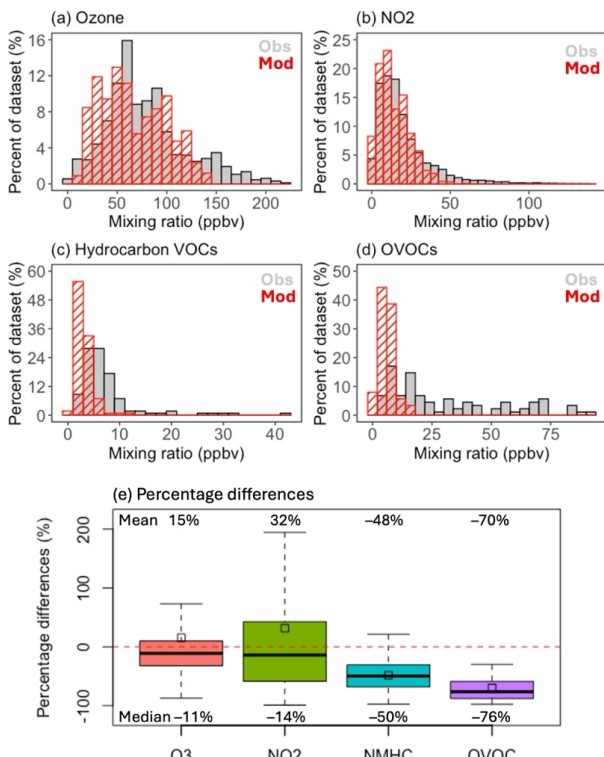

Figure 6. Frequency distribution between ship-observed (in grey) and modeled (in red)
concentrations for (a) ozone, (b) NO$_2$, (c) NMHCs, and (d) OVOCs. VOC comparisons are
conducted for equivalent species between 'selective observed' and 'modeled'. The percentage
differences for each species are summarized in (e).



We further break down these discrepancies into speciated differences in Figure 7. The results
indicate that, while certain species can be overestimated under specific conditions, such as
lumped xylene and other polyalkyl aromatics (XYLMN) and acetone (ACET) at land-based
measurements (Fig. 7b; Fig. S7), the majority of species are underestimated in the model, with
propane (PRPA) and ethyne (ETHY) showing the most significant underestimations. Propane is
underestimated by an average of 76% compared to shipborne measurements and 57% for land-
based measurements. Time series from land-based sites indicate that the underestimation mainly
occurs in winter and early spring (Fig. S4). Both ethane and propane are significantly affected by
leakage from oil and natural gas production and use (Ge et al., 2024; Guo et al., 2007), resulting
in a strong correlation between the two (Fig. S5). The model accurately captures such
correlation; however, ethane (ETHA) concentrations and variability are better simulated than
those of propane (Fig. S4). This underestimation of propane during colder months, while ethane
is reasonably simulated, has also been observed in Europe (Ge et al., 2024). This suggests that,
while the major emitting sectors for propane and ethane are well represented, the proportion
assigned to propane in the sector-specific emission profile may be insufficient.
Meanwhile, ethyne is underestimated by 73% and 88% relative to shipborne and land-based
measurements, respectively. Ethyne, ethene (ETH), and benzene (BENZENE) are recognized as
key tracers for combustion-related activities, particularly from vehicular and residential sources,
in emission inventories (von Schneidemesser et al., 2010; Lyu et al., 2016; Ho et al., 2009). This
is reflected in the strong correlations among these species simulated by the model (R ranging
from 0.7 to 1). However, observations show only moderate correlations in the ethene-to-ethyne
and benzene-to-ethyne relationships (R = 0.5; Fig. S5), suggesting the presence of regionally
unique sources of ethyne that may not be accounted for.
Speciated discrepancies for six OVOCs are also shown in Fig. 7, including formaldehyde
(FORM; –42% over water and –23% on land), acetaldehyde (ALD2; –57% over water and –58%
on land), aldehydes with three and more carbons (ALDX; –69% over water and –53% on land),
acetone (ACET; –65% over water and +99% on land), glyoxal (GLY; +30% over water), and
methylglyoxal (MGLY; –27% over water). While acetone at land-based sites and glyoxal over
coastal waters are overestimated on average, other OVOC species remain underestimated by the
model. Previous assessments of model performance in simulating OVOCs have been limited by
the scarcity of available measurements and species (Rowlinson et al., 2024; Ge et al., 2024; She
et al., 2024). Here, our unique capability to measure a series of OVOCs helps to address this gap.





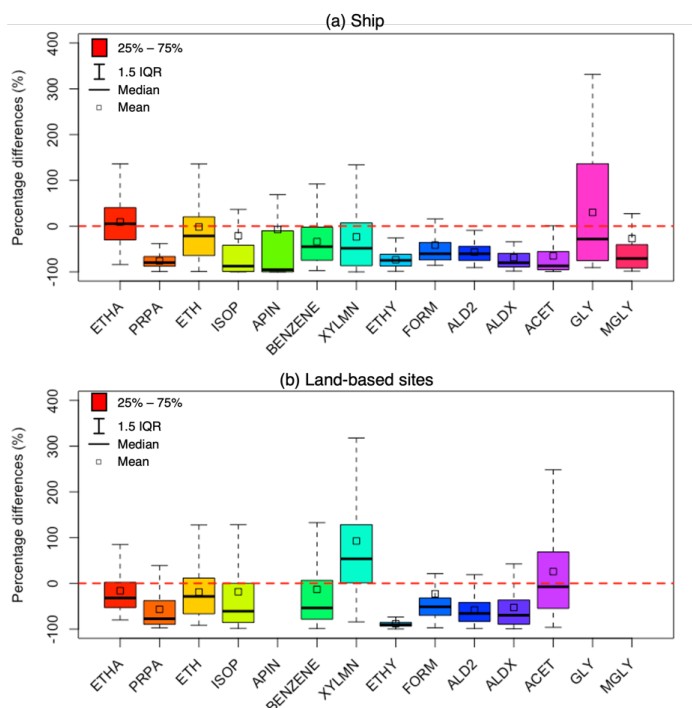

2    Figure 7. The percentage differences between the observation and model for individual VOC

3    species measured (a) on the ship and (b) at three land-based sites. Land-based measurements do

4    not detect alpha-pinene (APIN), glyoxal (GLY), and methylglyoxal (MGLY).



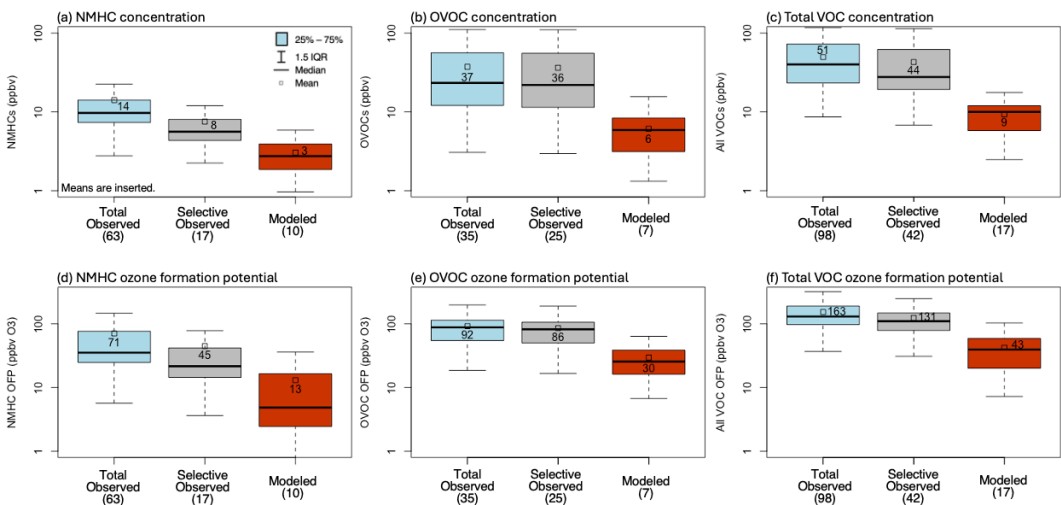

Figure 8. Observation–model comparison of concentration and ozone formation potential (OFP) of NMHCs and OVOCs. 'Total observed' includes all measured species. 'Selective observed' and 'modeled' refer to the equivalent species common to both observations and the model. Number in brackets indicates the number of species in each category. The y-axis is displayed on a logarithmic scale.

In the second comparison, we analyzed the gap between 'modeled' versus 'total observed' species (see Sect. 2.4 for details) to assess the overall representativeness of the modeled VOCs in relation to the full range of VOCs present in the ambient atmosphere. Figure 8 shows that the OVOC concentration of 37 ppbv accounts for 73% of the total VOC concentration and 56% of the total OFP, with NMHCs contributing a minor proportion. The total concentration of all measured VOCs of 51 ppbv can be broken down into three components when compared to the model (Fig. 8c). First, the concentrations successfully represented by the model account for 18% (9 ppbv), as shown in the 'modeled'. Second, model underestimation of equivalent species contributes 69% (35 ppbv = 44 ppbv – 9 ppbv), indicated by the differences between 'selective observed' and 'modeled'. Third, model-omitted portion account for 14% (7 ppbv = 51 ppbv – 44 ppbv), shown by the differences between 'total observed' and 'selective observed'. For OFP, the total measured is divided into 26% successful model representation, 54% model underestimation, and 20% model-omitted portion (Fig. 8f). This result in 82% and 74% differences between 'modeled' and 'total observed' concentrations and OPF, respectively, highlighting the missing VOC reactivity in the model.



**5. Role of VOCs in ozone pollution regulation**
To further examine how the model underrepresentation of VOCs affects simulated ozone
pollution regime, we use spaceborne tropospheric HCHO-to-NO$_2$ ratio (FNR) as an indicator to
infer regional ozone formation sensitivity (Wang Y. et al., 2023a; Jin et al., 2017; Goldberg et al.,
2022; Acdan et al., 2023). Tropospheric FNR over Hong Kong exhibits distinct seasonal
variation, with lower values during the cold months and higher values during warm months (Fig.
9c). According to a set of region- and season-specific FNR thresholds derived for the PRD
(Wang Y. et al., 2023a), we find that the cold months from November to April are VOC-limited,
while the warm months from May to October fall into a transitional regime sensitive to both
VOC and NO$_x$ (Fig. 9c). We acknowledge that this threshold should not be static in all scenarios,
and other studies have also derived FNR thresholds but for China as a whole (Ren et al., 2022; Li
et al, 2021; Wang et al., 2021). Our PRD region-specific thresholds from Wang Y. et al. (2023a)
fall within the ranges established by other independent studies. Yet, because our thresholds are
specifically tailored for our study region PRD, we place greater confidence in adopting them for
our study compared to more general national conditions presented in other studies. Among all
months, September shows the largest FNR (Fig. 9c) and the greatest monthly levels of ozone and
HCHO (Fig. 2), making it highly relevant for regulatory management. Therefore, we take
September 2022 as an example to illustrate the extent to which VOC underrepresentation in the
model can potentially affect the spatial ozone sensitivity diagnosis and thereby regulatory
measures in the following text.

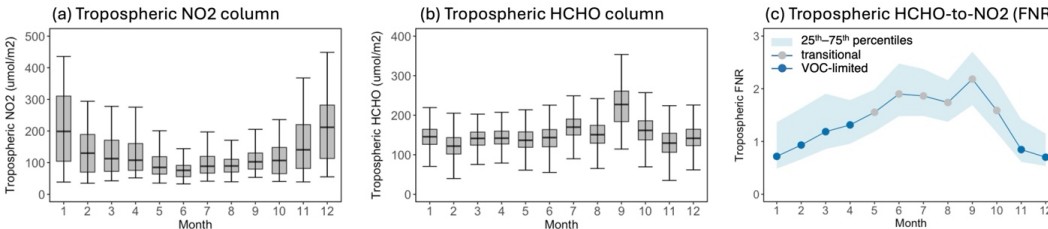

Figure 9. Monthly variation of TROPOMI tropospheric vertical column densities (VCDs) of (a)
NO$_2$ and (b) HCHO over the Hong Kong domain during 2021–2022. (c) The tropospheric
HCHO-to-NO$_2$ ratio (FNR) is calculated by dividing (a) by (b). The Pearl River Delta region-
specific FNR thresholds for differentiating ozone formation regimes are from Wang Y. et al.
(2023a): spring and summer [1.55, 3.05], autumn [1.45, 2.95], and winter [1.65, 3.15].



TROPOMI indicates elevated $NO_2$ and HCHO pollution in urban areas on both the eastern and
western flanks of the Pearl River Estuary (Fig. 10a, c). The Estuary also experiences high
pollution plumes, primarily influenced by shipping emissions over water and anthropogenic
activities from nearby urban clusters. Compared to TROPOMI, the model captures the general
spatial pattern (R = 0.8 for $NO_2$ and R = 0.6 for HCHO) but the magnitudes vary (Fig. 10b, d).
Due to background noise in spaceborne retrievals over extensive low pollution areas, which
significantly interferes with observation-model comparisons for the domain as a whole, the
domain mean or median does not well reflect differences in key polluted region. Therefore, we
overlay surface measurements of $NO_2$ and HCHO onto these urban areas and focus our
investigation on them. In this way, independent model-satellite and model-surface comparisons
can be used together and verify one another in understanding the differences between
observations and model in polluted regions. Our findings indicate that both surface and column
comparisons agree on $NO_2$ overestimation in highly polluted areas such as southern Guangzhou
(GZ), eastern Foshan (FS), and western Shenzhen (SZ), while showing underestimation in the
less polluted northern Guangzhou (Fig. 10b). HCHO demonstrates regional underestimation
(Fig. 10d). The median differences between surface observations and the model indicate a 25%
underestimation of $NO_2$ and a 29% underestimation of HCHO in the model.
The greater underestimation of HCHO compared to $NO_2$, consistent with findings from Sections
3–4,  results in the modeled FNR being 7% lower than surface observations (Fig. 10f). Similarly,
the model also simulates lower column FNRs than that observed by TROPOMI in polluted
regions (Fig. 10f). This discrepancy indicates that the modeled ozone formation is more sensitive
to VOCs than what surface measurements suggest in polluted regions. To probe into the ozone
formation regimes inferred from spaceborne FNR, we used the fall season FNR thresholds
tailored for the PRD from Wang Y. et al. (2023a) and identified a $NO_x$-limited ozone formation
regime in extensive suburban areas (FNR > 2.95; 91.7% of total domain grids), a transitional
regime in city clusters and busy shipping lanes of the eastern Pearl River Estuary (1.45 ≤ FNR ≤
2.95; 8.1% of total grids), and a VOC-limited regime on Hong Kong Island and Lantau (FNR <
1.45; 0.1% of total grids) (Fig. 10g). The model identifies a larger VOC-limited region,
accounting for 1.4% of total domain grids (Fig. 10h), suggesting that it overestimates the extent
of areas classified as VOC-limited regimes. This is partly attributed to inadequate representation
of VOCs within the model framework. We note that regimes can vary depending on the choice of
FNR thresholds. To address this, we tested various FNR thresholds derived from Wang Y. et al.
(2023a) and other independent studies, and all results agree that the model simulates more VOC-
limited regions than TROPOMI, despite differing regime classifications (Fig. S8; Fig. S9).




As models serve as an important tool for deriving ozone sensitivity and pollution control
strategies for regulatory agencies worldwide, their underrepresentation of VOCs results in a
greater sensitivity to VOCs in models than what is observed in ambient air. Regions identified as
transitional regime, which could benefit from coordinated emission reductions of both $NO_x$ and
VOCs as inferred from TROPOMI, are suggested by the model to achieve ozone reduction
primarily through VOC reductions alone (Fig. 10). This may lead to confusion for regulatory
agencies in developing effective ozone control strategies. At large, the underestimation of VOCs
in models has been noted in other Chinese cities (She et al., 2024; Wang S. et al., 2024), as well
as in Europe (Ge et al., 2024), North America (Chen et al., 2019), and globally (Rowlinson et al.,
2024). Community efforts are urged to integrate global VOC speciated measurements, with a
particular focus on OVOC components that have been less emphasized in model validation but
are poised to gain significance in the future (von Schneidemesser et al., 2023; Coggon et al.,
2021; Karl et al., 2018). From a modeling perspective, follow-up studies aimed at improving
model processes (e.g., emission, chemistry, etc.) will be crucial for advancing VOC-related
model development.





Figure 10. Spatial patterns of TROPOM-observed (left) and CMAQ-modeled (right) tropospheric vertical column densities (VCDs) of $NO_2$ and HCHO, the column HCHO-to-$NO_2$ ratio (FNR), and ozone formation regime in September 2022. Dots represent surface measurements and their model equivalents from September 4 and 5, 2022; the medians of observation-model differences are inserted. Dashed line boxes indicate the Hong Kong domain.





**6. Conclusions**
The HKEPD-HKUST field campaign conducted in 2021–2022 provided the first overwater
measurements of ozone and its precursors in Hong Kong coastal waters, complementing land-
based monitoring efforts. Notably, the campaign advances in the ability to measure a wide array
of VOC species, including a total of 98 species (63 MNHCs and 35 OVOCs) over water and 45
species (29 MNHCs and 16 OVOCs) on land. Such detailed speciation, particularly of
oxygenated compounds, is valuable for model evaluation, especially given the limited
availability of such measurements reported globally (Ge et al., 2024; von Schneidemesser et al.,
2023; She et al., 2024). This study leverages these field observations together with spaceborne
TROPOMI data to assess the regional VOC and ozone pollution dynamics, and, more
importantly, to diagnose VOC representation in a chemical transport model and its implication
on ozone pollution regulation.
Ozone levels in subtropical Hong Kong display a bimodal pattern, characterized by a minor peak
in spring and a significant peak in autumn. Oxygenated compounds, particularly HCHO, reach
their highest level in September, aligning with the peak monthly ozone levels. This underscores
the active photochemistry in autumn and its contribution to VOC and ozone pollution.
Comparing these observed features with the CMAQ model, we found that VOCs are typically
underestimated by 47%–48% for NMHCs and 45%–70% for OVOCs over land and water. A
detailed breakdown of the speciated comparison reveals the most significant underestimations in
propane (76% over water and 57% on land) and ethyne (73% over water and 88% on land).
Oxygenated compounds are also underestimated to varying degrees, such as formaldehyde (–
42% over water and –23% on land), acetaldehyde (–57% over water and –58% on land). These
differences between observations and models are referred to as the "underestimation" of
equivalent VOC species.
Through speciated analysis, we infer potential reasons for the underestimation of VOC to support
model development. Propane, primarily sourced from oil and natural gas leaks, and ethyne,
linked to combustion activities, are primary species whose ambient levels are mostly emission-
controlled. The model discrepancies suggest issues with emissions; increasing direct emissions,
such as addressing unaccounted sources for ethyne and adjusting the insufficient proportion
assigned to propane in the emission profile, may help mitigate the underestimation. In contrast,
oxygenated compounds have both primary sources and are generated as oxidation products from
other VOCs. This complexity makes it challenging to ascertain whether the underestimation
arises from missing primary emissions, inadequate secondary chemical production, or
overestimated chemical and deposition losses. Moving forward, there is a growing need to refine
the speciation profiles (Rowlinson et al., 2024; Ge et al., 2024) and unaccounted sources (She et



al., 2024; Travis et al., 2024; Beaudry et al., 2025) of VOC emissions, as well as to dissect a
more detailed breakdown of the chemical mechanisms for oxygenated species (Travis et al.,
2024; Gao et al., 2024) to improve model representation.
Apart from underestimation, another important aspect is that the model simulates only certain
species from the total measured VOCs. Specifically, only 28 out of the 45 VOC species observed
on land and 42 out of the 98 measured over water are represented in the model. These species
that are conventionally considered less important and thus not included in the model are defined
as model "omission". In our case, the model omits 17 species (38%) observed on land and 56
species (57%) observed over water, respectively. Together, both "omission" and
"underestimation" contribute to what we define as "VOC underrepresentation" for models. This
way, the total concentration of all measured VOCs of 51 ppbv over water can be broken down
into three factors when compared to the model: 18% successful model representation, 69%
model underestimation of equivalent species, and 14% model-omitted portion. Similarly for
OFP, the total measured is divided into 26% model representation, 54% model underestimation,
and 20% model omission. Among the three factors, underestimation is the most important;
addressing it is both impactful and feasible, making it a priority for model development. Omitted
species require further evidence to identify which specific species or groups are particularly
important yet overlooked in our current understanding.
The community acknowledges that previous modeling studies have primarily focused on
NMHCs, while OVOC assessments have been limited by the scarcity of available
measurements (Ge et al., 2024; von Schneidemesser et al., 2023; She et al., 2024). Our study
measured a wide array of OVOCs to help address this gap. We found that OVOCs account for
73% of the total measured concentration and 56% of the total OFP of all measured VOC species
over water. Despite their importance, OVOCs are underestimated to a greater extent than
NMHCs in the model. While emissions of short-chain NMHCs associated with fossil fuels and
combustion have notably decreased, future emission patterns are shifting towards increased
OVOC due to rising solvent use and consumer products (Beaudry et al., 2025; von
Schneidemesser et al., 2023; Coggon et al., 2021; Karl et al., 2018). Our OVOC assessment is a
valuable reference for improving OVOC representation in models and provides insights into how
model underrepresentation affects the future atmosphere amid evolving emission patterns.
The insufficient representation of VOCs in the model impacts policy-relevant ozone formation
sensitivity and pollution control strategies. We approach this using the FNR method, which is
widely used to diagnose ozone formation sensitivity (Wang Y. et al., 2023a; Jin et al., 2017;
Goldberg et al., 2022; Acdan et al., 2023). Both spaceborne TROPIMI and surface measurements
indicate lower FNRs simulated by the model than those observed in pollution hotspots in the



PRD region, implying that the simulated ozone formation may be overly sensitive to VOCs. As
models are critical tools for deriving ozone sensitivity and pollution control strategies for
regulatory agencies worldwide, the underrepresentation of VOCs could lead to skewed
understanding of ozone sensitivity and misguide effective control measures. This issue ripples
through broader geographical regions, as VOC underrepresentation has also been noted in other
Chinese cities (She et al., 2024; Wang W. et al., 2024), as well as in Europe (Ge et al., 2024),
North America (Chen et al., 2019), and globally (Rowlinson et al., 2024). From a boarder Earth
system perspective, such VOC underrepresentation may be associated with the OH reactivity
underestimation in models (Travis et al., 2024; Kim et al., 2022; Ferracci et al., 2018), which has
significant implications, not only for regional air quality through ozone and secondary organic
aerosols, but also for the lifetimes of greenhouses gases and global climate (Turner et al., 2019;
Zhao et al., 2025; Yang et al., 2025).
Overall, our study demonstrates the significant contribution of OVOCs in both the urban
environment of Hong Kong and its adjacent coastal waters, reveals the underrepresentation of
VOCs in models, and highlights the impact of this VOC underrepresentation on ozone pollution
regulations. Our findings serve as a valuable reference for understanding regional VOC and
ozone dynamics, advancing VOC-related model development, and fostering a synergistic
assessment of the broader impacts of underrepresented VOCs on air quality and climate from a
coherent Earth system perspective.



**1 Data availability**

WRF and CMAQ are open-source models and can be obtained from their developers at
https://github.com/wrf-model/WRF (last access: 6 June 2025) and
https://github.com/USEPA/CMAQ (last access: 6 June 2025), respectively. Model output data
and field measurements are available upon request.

**6 Competing interests**

The contact author has declared that neither they nor their co-authors have any competing
interests.

**9 Author contribution**

XL designed the study, conducted the analysis, and drafted the initial manuscript with guidance
from JCHF and ZW. YH performed the CMAQ simulations. ZW, YC, XF, YX, and YC provided
the shipborne OVOC data. DG and SH provided the shipborne NMHC data. ZN provided the
shipborne $NO_x$ and ozone data. JY and BC provided the HKUST supersite data. CL, YX, and TZ
provided ozone lidar data. CG and GB contributed to the TROPOMI analysis. All authors
participated in the preparation of the manuscript.

**16 Acknowledgments**

We appreciate the assistance of the Hong Kong Environmental Protection Department (HKEPD),
which provided some air quality data. This work was supported by the MOST
(2023YFC3709200) and the Research Grants Council of the Hong Kong Special Administrative
Region (Project No. C7041-21G, 16209022, 16211824, N_HKUST626/24). XL acknowledges
the Hong Kong University of Science and Technology (HKUST) Research Assistant Professor
(RAP) scheme and the Hong Kong Environment and Conservation Fund (Project No. 195/2024).
Any opinions, findings, conclusions or recommendations expressed in this material do not
necessarily reflect the views of the Government of the Hong Kong Special Administrative
Region and the Environment and Conservation Fund.

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
