# Peer review of "Abundance of volatile organic compounds and their role in ozone pollution management"

_EGUsphere, 2025_

## Referee Comment (RC1)

*Comments are Noted as "Pg X Line X" following preprint pages and line numbers.

General Comments:

- Typically, figures need to be placed before they are discussed, would suggest this rearrangement.

- Overall, in the model comparison to land and ship-based measurements, the ways in which both are compared to the model should be addressed in more detail as well as the limitations that each observational dataset has, as observations in land and over water are not uniform so their comparison to the model is not directly comparable or please state how they are in more detail.

Line Specific Comments:

Pg 2 Line 4: would specify what is occurring for the first time here – the integration of on land, shipborne, and spaceborne measurements in a study has already occurred in previous studies, so would ask for specifics on what is first time

Pg 4 Line 35: Caution the use of "first attempt" here. I think you should be more specific on to what was truly novel about this study, be it the ship-borne data captured in Southern China, the suite of species observed, think specifics would be welcomed

Pg 5 Line 2: remove 'also'

Pg 8 Section 2.2: Would be good to have that NO2 and HCHO observed by TROPOMI are used for proxies of Nox and VOCs, would be good to have this mentioned here and state more explanation is in Section 3.2

Pg 8 Section 2.3: Would be nice to inform which resolution was sampled to the TROPOMI overpass time and scale? All resolution? May need to explain choice and why downgrading or upgrading was performed and what information may be lost in the choice

Pg 11 Line 3: well captures – captures well?

Pg 11 Line 32: Would be nice to explain how the matching occurred here? Is it a TROPOMI average or a single TROPOMI pixel for 1 or more in situ measurement? Is there a domain considered? Specifics like this are welcomed

Pg 12 Line 11: would be careful comparing a surface measurement to a column, since they are not measuring the same thing, and maybe state that this is a relative comparison looking for trends more explicitly (if this is the case). Many try to use the column as a proxy for the surface, so would be clearer if this is what is occurring or if its just looking for trends

Pg 12 Line 11: Given the correlation between the surface and columns are quite low for this region, maybe other reasons for the low correlation are at play, like number of available surface stations compare the breadth of TROPOMI? Has this been considered?

Pg 12 Line15 – 18: Is having the cause of a local anthropogenic emitter is missing some causality for the disagreement, maybe it's the lower temperatures causing a less vertical mixing, etc.? Or it may be best to not speculate here?

Pg 12 Line 21: Where is the support for this correlation value? And the subsequent ones in the sentence? Related to a figure?

Pg 13 Line 4: same here? Where is the support for this? A figure?

Pg 13 Line 14: might be helpful to add if its land or ship or space-measure VOCs

Pg 13 Line 28- Pg 14 Line 10: This feel out of place here in Results. Could this paragraph could be incorporated into another section than here, but understand it was added for support for why HCHO can be used, but feels slightly out of place. Maybe just add line 28-30 as the concluding sentence and add the rest to the discussion?

Pg 14 Line 3: Do they really filter out areas with low NO2 or just focus on urban areas for its relation to large populations?

Pg 14 Figure 3: Difficult to view names in circle plot, is there a better way to show this? Maybe external legend on side or something?

Pg 15 Line 3: again would emphasize that this region is southern China

Pg 15 Line 18: ozone pool? A short explanation might be helpful here, like just adding a "ozone pool (i.e., X) simulated by …"

Pg 15 Line 30: Least – less?

Pg 15 Line 32: substantial positive or negative correlation?

Pg 16 Figure 4: I just wonder if there is a way to better display this data? Like reduce the latitudinal extent to see the data better?

Pg 17 Figure 5: Figure 5 is only discussed once in the paper and there is no discussion of its subplots, I wonder if more discussion of it could be included or if it can be condensed into a single plot?

Pg 17 Line 8: Specifics on which observations your using here would be helpful. And, information on how they are directly compare would be helpful too.

Pg 19 Line 6: Are propane (and the other species you directly compare) on land and on the ship measured using the same method? It might be helpful to identify the discrepancies between land and ship methods when having a direct comparison, as different measuring techniques can lead

to different measurements. Or state here again for the reader that they are measured in the same manner.

Pg 19 Line 12: Left out short name for Propane?

Pg 19 Line 12: No studies for China?

Pg 19 Line 13 – 15: I'm struggling to see this link to what was represented in the previous lines, I wonder if there is a better way to discuss this?

Pg 19 Line 17: Left out short name for Propane?

Pg 19: Line 20: struggling to see how this correlation related to the previous statement. Maybe more support in the specifics of how it relates can be added here?

Pg 19 Line 21: Land and ship observations? Might be helpful to add which observations for clarity

Pg 22 Line 13: any references for these studies?

Pg 23 Line 6: "background noise" what are you meaning here? Is this a point of magnitude or something else?

Pg 23 Line 9: Again may be helpful to share how the comparisons were done, in grid to point or area to point? And also again would ask for clarification on if these are relative comparisons?

Pg 23 Line 18: Which HCHO? TROPOMI or Model?

Pg 26 Line 11: would add what chemical transport model here for clarity, just as you do for spaceborne and TROPOMI

Pg 27 Line 32: May also be beneficial to add the limitations of the FNR? Souri et al, 2025 (https://acp.copernicus.org/articles/25/2061/2025/) has good descriptions on the limitations of the FNR

Pg 28 Line 28: model discrepancies in propane? Discrepancies in what? Specifics are desired

---

## Author Response (AR1)

**Author Responses to Reviewers' Comments on "Abundance of volatile organic compounds and their role in ozone pollution management: Evidence from multi-platform observations and model representations during the 2021–2022 field campaign in Hong Kong" by Liu et al. (MS No.: egusphere-2025-3227)**

We would like to thank the reviewers for the thoughtful and insightful comments. The manuscript has been revised accordingly, and our point-by-point responses are provided below. Reviewers' comments are in black font, our replies in blue, and our new/modified text is highlighted in **bold**. Line numbers in the responses correspond to the revised manuscript.

**Response to Reviewer #1**

*Comments are Noted as "Pg X Line X" following preprint pages and line numbers.
We thank the reviewer for the valuable comments, which have improved the manuscript.

General Comments:
- Typically, figures need to be placed before they are discussed, would suggest this rearrangement.
Response: Rearranged as suggested.

- Overall, in the model comparison to land and ship-based measurements, the ways in which both are compared to the model should be addressed in more detail as well as the limitations that each observational dataset has, as observations in land and over water are not uniform so their comparison to the model is not directly comparable or please state how they are in more detail.
Response: We have now added more descriptions on the observation–model comparisons and limitations.

P11 L25: "**In all observation–model comparisons, we first identified the original resolution of each observation. Given that the model results have a minimum temporal resolution of hourly and lack sub-hour variability, we used hourly averaged model concentrations for observations at hourly resolution or finer. For daily observations, we averaged the corresponding hourly model outputs to obtain the modeled daily values.**"

P6 L32: "**In summary, different sampling and analytical methods are used to obtain measurements of NMHCs and OVOCs. Both land and ship measurements used offline techniques for OVOCs; however, the UHPLC-MS employed in shipborne measurements was capable of detecting a broader range of OVOCs than the HPLC-UV used at land-based HKEPD sites (Table 1). Additionally, the offline GC-MSD/ECD/FID used in shipborne measurements also identified more NMHCs than the online GC-PID/FID used at land-based HKEPD sites. Quantifying intra-instrumental differences for the same species at the same location and time poses practical challenges. Readers should be aware of the uncertainties related to these differences.** The observational datasets used in this study are sourced from peer-reviewed publications that underwent a thorough quality assurance and quality control (QA/QC) process; readers interested in detailed QA/QC methodologies are recommended to refer to the Supplement Text or the original publications listed above."

Line Specific Comments:
Pg 2 Line 4: would specify what is occurring for the first time here – the integration of on land, shipborne, and spaceborne measurements in a study has already occurred in previous studies, so would ask for specifics on what is first time

Response: To our knowledge, this study marks the first shipborne measurements of air pollutants over Hong Kong waters, which enables the first integration of multi-source observations from shipborne, on-land, and spaceborne platforms to examine region air pollution in the PRD region. We have now clarified it as '**the first attempt to integrate multi-source observations from on-land, shipborne, and spaceborne platforms with chemical transport modeling to examine VOC and ozone pollution in the PRD region of South China**', as specified in P4 L31. We removed the 'first time' expression from the abstract (P2 L4) due to space constraints.

If the reviewer could kindly provide references for any previous work that has integrated these three types of measurements in the PRD region, we would be happy to revise our statements accordingly.

Pg 4 Line 35: Caution the use of "first attempt" here. I think you should be more specific on to what was truly novel about this study, be it the ship-borne data captured in Southern China, the suite of species observed, think specifics would be welcomed
Response: Please refer to our response to the above comment.

Pg 5 Line 2: remove 'also'
Response: Removed as suggested.

Pg 8 Section 2.2: Would be good to have that NO2 and HCHO observed by TROPOMI are used for proxies of Nox and VOCs, would be good to have this mentioned here and state more explanation is in Section 3.2
Response: We have now added the following statement in P9 L14: '**The tropospheric HCHO and NO2 columns can serve as proxies for VOCs and NOx, respectively, and thus their ratio (HCHO-to-NO2) indicates ozone formation sensitivity to changes in VOCs and NOx (Jin et al., 2017; Goldberg et al., 2022); further details can be found in Sect. 3.2.**'

Pg 8 Section 2.3: Would be nice to inform which resolution was sampled to the TROPOMI overpass time and scale? All resolution? May need to explain choice and why downgrading or upgrading was performed and what information may be lost in the choice
Response: We have now added corresponding content to clarify this point.

P9 L10: "This study used TROPOMI Product Algorithm Laboratory (PAL) tropospheric vertical column densities (VCDs) for a consistent reprocessed data product over 2021–2022 **at ~2 km × 2 km resolution** …"

P10 L32: "**Here, we opted to use model simulations at 3 km resolution for comparison with TROPOMI products at 2 km resolution, as these represent the closest match in spatial resolution in our study. This choice also allows us to highlight the differences between urban and suburban areas in Fig. 10, which a 1km model outputs would not adequately cover. Yet this choice may result in the loss of some fine-scale features simulated at 1 km resolution.**"

Pg 11 Line 3: well captures – captures well?
Response: Revised to 'captures well' as suggested.

Pg 11 Line 32: Would be nice to explain how the matching occurred here? Is it a TROPOMI average or a single TROPOMI pixel for 1 or more in situ measurement? Is there a domain considered? Specifics like this are welcomed
Response: We have now revised the sentence for better clarity. P12 L11: "This seasonal pattern of surface NO2 correlates well with the tropospheric VCDs of NO2 measured by the spaceborne TROPOMI (R =

0.9; Fig. S2) **and averaged over the Hong Kong domain (see domain coverage in Fig.10)**, which show monthly differences relative to the annual mean ranging from −20% in JJA to +60% in DJF".

Pg 12 Line 11: would be careful comparing a surface measurement to a column, since they are not measuring the same thing, and maybe state that this is a relative comparison looking for trends more explicitly (if this is the case). Many try to use the column as a proxy for the surface, so would be clearer if this is what is occurring or if its just looking for trends
Response: We have now clarified it as "**We further examined spaceborne tropospheric columns specifically to identify trends, and** found that …" in P13 L20.

Pg 12 Line 11: Given the correlation between the surface and columns are quite low for this region, maybe other reasons for the low correlation are at play, like number of available surface stations compare the breadth of TROPOMI? Has this been considered?
Response: Thank you for this helpful suggestion. The reviewer's speculation on the number of available surface stations compared to the breadth of TROPOMI makes sense. In addition, we believe several other factors can also be attributed. Decoupling between column and surface HCHO has been observed in many places, especially in coastal regions with complex land-sea interactions, as noted in Souri et al. (2023). We feel that a comprehensive discussion of the column-surface decoupling of HCHO would require a separate paragraph to address multiple reasons, which may drift away from the focus of this section. Therefore, we have included a concise sentence in the main text to inform readers on this column-surface decoupling while omitting the detailed reason behind, as shown below.

P13 L24: "**In general, we do not observe a strong correlation in monthly trends between column and surface HCHO; this column–surface decoupling is also noted in other locations particularly in coastal regions with intense land–sea interactions (Souri et al., 2023), similar to our study area.**"

Reference:
Souri, A. H., Kumar, R., Chong, H., Golbazi, M., Knowland, K. E., Geddes, J., and Johnson, M. S.: Decoupling in the vertical shape of HCHO during a sea breeze event: The effect on trace gas satellite retrievals and column-to-surface translation, Atmos. Environ., 309, 119929, 2023.

Pg 12 Line15 – 18: Is having the cause of a local anthropogenic emitter is missing some causality for the disagreement, maybe it's the lower temperatures causing a less vertical mixing, etc.? Or it may be best to not speculate here?
Response: We agree with the reviewer that speculation should be avoided in this context and therefore have removed the statement accordingly.

P13 L23: "In contrast, the January peak is visible only at the surface but not in tropospheric columns (Fig. S2). "

Pg 12 Line 21: Where is the support for this correlation value? And the subsequent ones in the sentence? Related to a figure?
Response: The correlations are derived from the data plotted in Figure 2, but in different sub-panels. We have now added a separate figure to display these correlations in Figure S5, and referenced it in the main text "A strong relationship exists between HCHO and ozone, as indicated by a strong correlation (R = 0.7; **Fig. S5**) between the observed monthly variability of the two" in P13 L30.

[Figure]

Figure S5. Correlation between monthly variability of ozone and VOCs at land-based sites.

Pg 13 Line 4: same here? Where is the support for this? A figure?
Response: Same as above.

Pg 13 Line 14: might be helpful to add if its land or ship or space-measure VOCs
Response: We have now clarified it as "First, we examine the relative contributions of HCHO to all measured VOCs at these **land-based** locations."

Pg 13 Line 28- Pg 14 Line 10: This feel out of place here in Results. Could this paragraph could be incorporated into another section than here, but understand it was added for support for why HCHO can be used, but feels slightly out of place. Maybe just add line 28-30 as the concluding sentence and add the rest to the discussion?
Response: Thank you for this suggestion. In this manuscript, we have combined the Results and the Discussion sections into Section 3-5, labeled according to their respective scientific content. In each section, we typically present the results followed by some discussion. As the reviewer recommended, this paragraph is included in Discussion Section.

Pg 14 Line 3: Do they really filter out areas with low NO2 or just focus on urban areas for its relation to large populations?
Response: We intended to convey that she focused solely on urban areas and filtered out low NO2 regions, as stated: 'We first assess if space-based HCHO/NO2 captures the nonlinearity of O3 chemistry by matching daily OMI observation with ground-based O3 measurements over polluted areas' and 'We only include sites over polluted regions (defined as long-term average OMI $\Omega NO_2 > 1.5 \times 10^{15}$ molecules/cm2)' in Jin et al. (2020).

For better clarity, we have now revised the expression to be more straightforward as the reviewer suggested: 'Studies that use the FNR to identify ozone formation regimes tend to  **focus on urban areas** with high $NO_2$ levels (Jin et al., 2020)…'

Pg 14 Figure 3: Difficult to view names in circle plot, is there a better way to show this? Maybe external legend on side or something?
Response: Figure 3 has been revised as suggested.

[Figure]

Figure 3. Contribution of individual VOCs to (a, d, g) the concentration and (b, e, h) the ozone formation potential (OFP) of the total observed VOCs at three land-based sites in 2021. (c, f, i) Time series of HCHO and total observed VOCs with their correlation coefficient inserted.

Pg 15 Line 3: again would emphasize that this region is southern China
Response: Revised as suggested.

Pg 15 Line 18: ozone pool? A short explanation might be helpful here, like just adding a "ozone pool (i.e., X) simulated by …"
Response: Revised to be 'ozone **reservoir**'.

Pg 15 Line 30: Least – less?
Response: Revised as suggested.

Pg 15 Line 32: substantial positive or negative correlation?
Response: Revised to be 'substantial **positive** correlation'.

Pg 16 Figure 4: I just wonder if there is a way to better display this data? Like reduce the latitudinal extent to see the data better?
Response: Revised as suggested.

[Figure]

Figure 4. Mobile ship measurements of ozone (top), NO2 (middle) and Ox (bottom), along with their modeled equivalents. Percentage differences are given as (model–observation)/observation.

Pg 17 Figure 5: Figure 5 is only discussed once in the paper and there is no discussion of its subplots, I wonder if more discussion of it could be included or if it can be condensed into a single plot?
Response: We agree with the reviewer and have condensed Figure 5 as suggested.

[Figure]

Figure 5. Ozone diurnal variations recorded during westward ship cruises and at two land-based sites (Tangjia and Tung Chung; locations shown in Fig. 1). The mean water-land ozone gradient (ΔO3) is calculated as shipborne ozone minus land-based measurements from 9 AM to 2 PM.

Pg 17 Line 8: Specifics on which observations your using here would be helpful. And, information on how they are directly compare would be helpful too.
Response: We have now specified it in P18 L7: "**Using shipborne measurements**, two types of observation–model comparisons were conducted for VOCs ...".

For better section arrangement, detailed information on how they are compared is provided in Section 2.4 P11 L17: "To evaluate the model's representation of VOCs, we conducted two types of observation–model comparisons. **First, we performed an equivalent comparison between 'modeled' and 'selective observed' species to assess model bias in simulating the concentrations of VOC species that are common to both the observations and the model.** This helps identify discrepancies in the model's ability to capture key species that significantly influence atmospheric chemistry, indicating either underestimation or overestimation. **Second, we compared 'modeled' to 'total observed' species to evaluate the overall representativeness of the modeled VOCs in relation to the ambient measurable VOC abundance.** This comparison provides insight into the model's ability to reflect the full range of VOCs present in the atmosphere. **In all observation–model comparisons, we first identified the original resolution of each observation. Given that the model results have a minimum temporal**

**resolution of hourly and lack sub-hour variability, we used hourly averaged model concentrations for observations at hourly resolution or finer. For daily observations, we averaged the corresponding hourly model outputs to obtain the modeled daily values.".**

Pg 19 Line 6: Are propane (and the other species you directly compare) on land and on the ship measured using the same method? It might be helpful to identify the discrepancies between land and ship methods when having a direct comparison, as different measuring techniques can lead to different measurements. Or state here again for the reader that they are measured in the same manner.
Response: We have now added more descriptions on measuring techniques between land and ship.

P6 L32: "**In summary, different sampling and analytical methods are used to obtain measurements of NMHCs and OVOCs. Both land and ship measurements used offline techniques for OVOCs; however, the UHPLC-MS employed in shipborne measurements was capable of detecting a broader range of OVOCs than the HPLC-UV used at land-based HKEPD sites (Table 1). Additionally, the offline GC-MSD/ECD/FID used in shipborne measurements also identified more NMHCs than the online GC-PID/FID used at land-based HKEPD sites. Quantifying intra-instrumental differences for the same species at the same location and time poses practical challenges. Readers should be aware of the uncertainties related to these differences.** The observational datasets used in this study are sourced from peer-reviewed publications that underwent a thorough quality assurance and quality control (QA/QC) process; readers interested in detailed QA/QC methodologies are recommended to refer to the Supplement Text or the original publications listed above."

Pg 19 Line 12: Left out short name for Propane?
Response: The short name for propane is mentioned at its first appearance in P19 L15.

Pg 19 Line 12: No studies for China?
Response: We have now added additional references and noticed that this finding is evident in many more geographical regions, including Asia, Europe, North America, etc.. Therefore, we now revised it to state: "This underestimation of propane during colder months, while ethane is reasonably simulated, has also been observed  **in various regions around the world** (Ge et al., 2024**; Rowlinson et al., 2024; Adedeji et al., 2023**)" in P20 L7.

Pg 19 Line 13 – 15: I'm struggling to see this link to what was represented in the previous lines, I wonder if there is a better way to discuss this?
Response: We have now revise it to be: "This underestimation of propane, while ethane is reasonably simulated, has also been observed in various regions around the world (Ge et al., 2024; Rowlinson et al., 2024; Adedeji et al., 2023). **Consequently, the model exhibits lower propane-to-ethane ratios compared to the measurements, suggesting the presence of unaccounted emission sources for propane**" in P20 L8.

Pg 19 Line 17: Left out short name for Propane?
Response: The short names for propane and ethyne are mentioned at their first appearances in P19 L15.

Pg 19: Line 20: struggling to see how this correlation related to the previous statement. Maybe more support in the specifics of how it relates can be added here?
Response: We have now revised it for better clarify: "Ethyne, ethene (ETH), and benzene (BENZENE) are recognized as key tracers for combustion-related activities, particularly from vehicular and residential

sources, in emission inventories (von Schneidemesser et al., 2010; Lyu et al., 2016; Ho et al., 2009). This is reflected in the strong correlations among these species simulated by the model (R ranging from 0.7 to 1)**, driven by the aforementioned emission inventories**" in P20 L12.

Pg 19 Line 21: Land and ship observations? Might be helpful to add which observations for clarity

Response: Revised to be "**land-based** observations" as suggested.

Pg 22 Line 13: any references for these studies?

Response: For better clarify, we have now revised the sentence in P22 L19 to be: "We acknowledge that this threshold should not be static in all scenarios, and other studies have also derived FNR thresholds but for China as a whole (Ren et al., 2022; Li et al, 2021; Wang et al., 2021). Our PRD region-specific thresholds from Wang Y. et al. (2023a) fall within the ranges established **by the aforementioned independent studies but for China as a whole.**".

Pg 23 Line 6: "background noise" what are you meaning here? Is this a point of magnitude or something else?

Response: We have now revised the expression in P24 L7 to be: "**Due to the prevalence of low values in less polluted suburban areas,** which significantly complicates observation-model comparisons for the domain as a whole, the domain mean or median does not well reflect differences in key polluted regions…".

Pg 23 Line 9: Again may be helpful to share how the comparisons were done, in grid to point or area to point? And also again would ask for clarification on if these are relative comparisons?

Response: We have now revised it for better clarity in P24 L9.

"Therefore, we overlay surface measurements of NO2 and HCHO onto these urban areas and **identify the nearest model grid to perform observation-model comparisons**. In this way, independent model-satellite and model-surface comparisons can be used together and verify one another in understanding the differences between observations and model in polluted regions. **It is important to note that we do not directly compare surface measurements with column measurements; instead, we use these two sources of observations to conduct separate comparisons with the model, aiming to identify observation-model differences.**"

Pg 23 Line 18: Which HCHO? TROPOMI or Model?

Response: Revised to be "The greater **model** underestimation of HCHO compared to NO2" in P24 L22.

Pg 26 Line 11: would add what chemical transport model here for clarity, just as you do for spaceborne and TROPOMI

Response: Revised as suggested in P25 L26.

Pg 27 Line 32: May also be beneficial to add the limitations of the FNR? Souri et al, 2025 (https://acp.copernicus.org/articles/25/2061/2025/) has good descriptions on the limitations of the FNR

Response: We have now added the limitation of FNR method, as suggested.

P27 L16: "The insufficient representation of VOCs in the model impacts policy-relevant ozone formation sensitivity and pollution control strategies. We approach this using the widely used FNR method for diagnosing ozone formation sensitivity, **while simultaneously acknowledging that several limitations exist for this method, including the inherent limitations in understanding the radical termination in the HOx–ROx cycle,  the challenges associated with converting the column vertical density to near-surface concentrations, the spatial representativity of large satellite pixels, and retrieval errors (Souri et al., 2025).**".

Pg 28 Line 28: model discrepancies in propane? Discrepancies in what? Specifics are desired

Response: We have now revised it for better clarity: "The model discrepancies **in propane** …".

**Response to Reviewer #2**

This manuscript integrates on-land, shipborne, and spaceborne measurements from a field campaign in Hong Kong during 2021-2022, analyzing 45-98 VOC species over land and water. This study presents OVOCs dominate total VOC concentration and OFP, highlighting their critical role in ozone chemistry. The CMAQ model significantly underestimates both OVOCs and NMHCs, and omits 17-56 measured species entirely. Model underrepresentation of VOCs leads to overestimated ozone sensitivity to VOCs, potentially misguiding pollution control strategies. The study integrates TROPOMI satellite data with surface measurements to evaluate spatial and temporal patterns of ozone precursors and diagnose model performance. The study provides valuable empirical evidence of VOC underrepresentation in models and underscores the growing importance of OVOCs in urban ozone formation. However, the QAQC for the VOC measurement is not comprehensive enough, especially for the OVOC species (e.g. formaldehyde). There are a few details that should be addressed, notably pay close attention to the readability of most figure needs to be improved throughout the manuscript.

We would like to thank the reviewer for the very thoughtful and insightful comments, which have improved the manuscript.

General comments

Page 4 Line 17: What is the difference between solvent use and consumer products and volatile chemical product (line 30, page 3). The literature cited here is mostly based on simulation studies conducted in the United States. Is there any research specifically targeting China? In fact, many studies have proposed that OVOCs emitted from VCP are the key to improving simulation. It is suggested to compare and analyze the OVOCs measured in this study with VCP emission related research.

Response:

(1) Solvent use primarily refers to coatings, adhesives, and similar products used in industrial and architectural applications including vehicles, motorcycles, vessels, containers, electronic equipment manufacturing; these are normally regarded as industrial volatile chemical products (VCPs) (Wang S. et al., 2024). Consumer products, as another emerging category of VCPs, primarily refer to personal care products, cleaning agents, pesticides, coatings, adhesives, and more (Wang Y. et al., 2024; Seltzer et al., 2021).

(2) We have now included studies specifically focusing on China and the PRD in P4 L17: '… and consumer products (**Wang S. et al., 2024; Wang Y. et al., 2024;** Coggon et al., 2021; Karl et al., 2018).'.

(3) The primary objectives of this study are to "quantify VOC representation in the CMAQ chemical transport model, and assess the impact of such VOC representation on the simulated ozone formation sensitivity", as outlined in the last paragraph of the Introduction section. We agree with the reviewer that exploring whether addressing OVOC emissions from VCPs can help reduce the observation-model bias identified in our study is a meaningful and interesting topic. However, given our current content layout and logic flow, along with the limited studies on local VCP emissions in Hong Kong, it is less straightforward to accommodate such analysis in the manuscript. Instead, we address this important point in the Conclusion section as a future research direction.

P26 L13: "… oxygenated compounds have both primary sources and are generated as oxidation products from other VOCs. This complexity makes it challenging to ascertain whether the underestimation arises from missing primary emissions, inadequate secondary chemical production, or overestimated chemical and deposition losses. **Many studies emphasize that addressing OVOC emissions from VCPs is crucial for improving model simulations (Wang S. et al., 2024; Wang Y. et al., 2024; Coggon et al., 2021; Karl et al., 2018).**".

Page 5 line 18-24: This study emphasizes the importance of OVOCs, but the measurement of OVOCs only uses offline sampling. Is this sampling done every day? How to ensure that the concentration of OVOCs data collected offline is at the same level as that of NMHC data collected online, or how to carry out quality control work? I don't think it's enough to simply add up two measurement methods to determine the importance of OVOCs. Please provide evidence of the reliability of this data. Regarding the reliability of the measurement methods, at least it needs to be given in the SI.

Response:

(1) We have now provided additional details on the sampling. P5 L24: "…daily OVOC samples were collected using 2,4-dinitrophenylhydrazine (2,4-DNPH) cartridges **every 2–7 days**  …"

(2) We have now incorporated QA/QC in the SI and referenced it in the main text. P7 L3: "**The observational datasets used in this study are sourced from peer-reviewed publications that underwent a thorough quality assurance and quality control (QA/QC) process; readers interested in detailed QA/QC methodologies are recommended to refer to the Supplement Text or the original publications listed above.**"

(3) We have now provided clarification on the summation of NMHCs and OVOCs to obtain the total observed VOCs.

P5 L27: "**To capture the day-to-day variation of total observed VOCs discussed in Sect. 3.2, we first identified dates with both NMHC and OVOC measurements; for each of these days, we averaged the hourly NMHC values to obtain a daily mean and combined this with the daily OVOC value to calculate the total daily concentration of all measured VOCs.**"

We have ensured that determining the contributions of VOC groups measured by different methods is a practical and accepted approach within the community, as in Wu et al. (2020) published in ACP.

[Figure]

Figure R1. The concentration percentage of each category of VOCs measured by different instruments, adopted from Figure 5 in Wu et al. (2020).

Reference: Wu, C., Wang, C., Wang, S., Wang, W., Yuan, B., Qi, J., Wang, B., Wang, H., Wang, C., Song, W., Wang, X., Hu, W., Lou, S., Ye, C., Peng, Y., Wang, Z., Huangfu, Y., Xie, Y., Zhu, M., Zheng, J., Wang, X., Jiang, B., Zhang, Z., and Shao, M.: Measurement report: Important contributions of oxygenated compounds to emissions and chemistry of volatile organic compounds in urban air, Atmos. Chem. Phys., 20, 14769–14785, 2020.

Page 5 Line26: I think HCHO should be formaldehyde here, and it would be better not to directly use its chemical formula as a name in the main text or explain it when first used. Is this "40 sites snapshot" only analyzing the concentration of formaldehyde? Has the reliability of formaldehyde concentration been

demonstrated by comparing other OVOC species sampled by DNPH cartridges? Why are the two HPLC selections different, and does this have an impact on the analysis results? Meanwhile, what other work were taken to ensure the representativeness of this 40 sites snapshot?

Response:

(1) HCHO acronym is now defined at first appearance.

(2) Gridded sampling has other VOCs measured, but here we only introduce HCHO as surface measurements to compare with spaceborne HCHO column measurements in Section 5. The QA/QC for DNPH cartridges is now provided in the Supplementary Text and referenced in the main text.

P7 L3: "**The observational datasets used in this study are sourced from peer-reviewed publications that underwent a thorough quality assurance and quality control (QA/QC) process; readers interested in detailed QA/QC methodologies are recommended to refer to the Supplement Text or the original publications listed below.**"

(3) The differences between the two land-based HPLCs do not affect our analysis, as we did not directly compare land-based HKEPD measurements with gridded sampling over the PRD region.

HPLC selections differ between land-based HKEPD sites and shipborne measurements: HPLC-UV is used for regulatory sites in accordance with HKEPD requirements, while UHPLC-MS is used for shipborne measurements conducted by HKUST. UHPLC-MS can detect a broader range of VOC species than HPLC-UV. We have now addressed it as below.

P6 L33: "**Both land and ship measurements used offline techniques for OVOCs; however, the UHPLC-MS employed in shipborne measurements was capable of detecting a broader range of OVOCs than the HPLC-UV used at land-based HKEPD sites (Table 1).**"

(4) We have now added more description of the representativeness of this 40 sites snapshot.

P6 L1: "**To ensure representativeness for the PRD regional conditions, the area was mapped into a 200 km × 200 km domain, divided into 100 grid cells of 20 km × 20 km each; measurement sites were scattered within these grid cells and located at least 50 m away from local pollution sources to capture well-mixed air within each grid cell (Mo et al., 2023).**"

Page 5-6 Section 2.1.1-2.1.2: I think the description of the instrument here is pointless. Suggest summarizing and rewriting these sentences. In the methodology section, it should not only list the instruments used, but also express more about the measurement ability of VOCs species in this study, the differences between different measurement methods, and the addition of QAQC explanations to verify the reliability of the data used in this study. In the two types of measurements, the quality control process for species measured by GC-MS needs to be provided (species in Table 1), at least in the SI section, and cannot be solely based on references. Additionally, the molecular formula of ethylbenzene and m/p-xylene (C8H10) is incorrect in Table 1.

Response:

(1) The measurement capabilities and differences have now been added in Method.

P6 L32: "**In summary, different sampling and analytical methods are used to obtain measurements of NMHCs and OVOCs. Both land and ship measurements used offline techniques for OVOCs; however, the UHPLC-MS employed in shipborne measurements was capable of detecting a broader range of OVOCs than the HPLC-UV used at land-based HKEPD sites (Table 1). Additionally, the**

**offline GC-MSD/ECD/FID used in shipborne measurements also identified more NMHCs than the online GC-PID/FID used at land-based HKEPD sites.**"

(2) We have now incorporated QA/QC in the SI and referenced it in the main text. P7 L3: "**The observational datasets used in this study are sourced from peer-reviewed publications that underwent a thorough quality assurance and quality control (QA/QC) process; readers interested in detailed QA/QC methodologies are recommended to refer to the Supplement Text or the original publications listed below.**"

(3) The molecular formulas in Table 1 are now corrected.

Page 10 Section 2.4: How is the species time resolution of NO2 and VOCs (especially formaldehyde) unified among different measurement methods in this study? Is there significant uncertainty, especially when conducting FNR value analysis?
Response: Unified temporal resolutions were applied. We have revised the text for better clarity.

P6 L7: "At each site, two HCHO samples were collected: one in the morning (6:00–10:00) and the other in the afternoon (12:00–16:00) ... For each HCHO sampling site, **we matched the corresponding afternoon mean NO2** from the nearest station of the China National Environmental Monitoring Center (CNEMC) **during the same period (12:00–16:00)** to calculate the surface HCHO-to-NO2 ratio for comparison with spaceborne ratio."

Page13 Line14-27: When conducting OFP analysis, it is suggested to propose the MIR value of formaldehyde species to highlight that formaldehyde also contributes significantly to OFP at high concentrations, distinguishing it from other species that only contribute high concentrations or only contribute high OFP. Meanwhile, it is difficult to see from the time series in Figure 3 that the correlation performance between formaldehyde and measured total VOC reaches 0.72-0.85. It is recommended to add scatter plots and linear relationship fitting graphs.
Response: Following reviewer's suggestion, we have now included this statement. P15 L15: "**It is worth noting that HCHO is a significant contributor to both VOC concentration and OPF due to its high levels and large MIR value (9.46 g O3/g VOC), distinguishing it from other species that may either only show high concentrations or only contribute significantly to OFP, particularly at the Tung Chung and Hok Tsui sites.**"

Scatter plots have been added as Figure S7, and are referenced in the main text as in '… we examine the temporal correlation between HCHO and total measured VOCs and find strong correlations at all sites: R=0.81 at Mong Kok (Fig. 3c**; Fig. S7**), R=0.85 at Tung Chung (Fig. 3f**; Fig. S7**), and R=0.72 at Hok Tsui (Fig. 3i**; Fig. S7**)…'.

[Figure]

Figure S7. Correlation between HCHO and total observed VOCs at three land-based sites.

Page 13 Line 28-Page 14 Line 10: I think formaldehyde as a reliable indicator of ambient VOCs is not convincing enough. This study only selected 3 sites tested in Hong Kong, and the number of VOCs species measured is limited (45 species). It is suggested to add more comparisons with existing literature, such as adding test results from Guangzhou/Shenzhen urban areas in the PRD region, to verify the regional representativeness of this statement.

Response: Hong Kong only has three sites with regular OVOC sampling; we selected all of them. VOCs are not routinely measured as criteria pollutants at China National Environmental Monitoring Center (CNEMC) sites in the PRD. There is no consistent year-long record of VOC observations for assessment. To address the regional representativeness raised by the reviewer, we used data from a 2-day campaign conducted in the PRD.

We have now added the content in P15 L6: "**Taking a case study of the PRD region for example, we find that HCHO contributes to 15%–30% of the total measured VOC concentrations from ~ 50 species; this percentage declines to 8%–21% when using another dataset that includes ~80 species (Fig. S6). Notably, these percentage contributions in the PRD region are similar to those observed in Hong Kong, confirming the regional representativeness of HCHO's contribution.**".

[Figure]

Figure S6. The contribution of HCHO to the total concentration of all observed VOCs. OVOCs were measured using (a) HPLC-UV for 16 species and (b) UHLPC-MS for 43 species. NMHCs were measured using the same method in both (a) and (b), totaling 38 species. Overall, there are 54 observed VOC species in (a) and 81 in (b).

Page 14 Figure 3: The overlapping species names here are difficult to distinguish. It is recommended to only display the species with higher contributions, and other species should be presented separately in the form of legends or by category. It would be better if NMHC and OVOCs could be clearly distinguished. At the same time, it is necessary to unify the naming convention for formaldehyde and HCHO on the figure. The naming of species should be standardized, such as for aldehydes 3C+ without any explanation and non-standard, at least using expressions like C3-C6 aldehydes.

In addition, all VOCs concentrations are daily average concentrations, which are easily affected by nearby emission sources or human factors, resulting in significant data deviations. Has this part been checked and corrected accordingly?

Response: Figure 3 has been revised as suggested. We have now placed the five OVOC species at the top of the legend to help readers more easily distinguish them from the NMHCs listed at the bottom.

[Figure]

Figure 3. Contribution of individual VOCs to (a, d, g) the concentration and (b, e, h) the ozone formation potential (OFP) of the total observed VOCs at three land-based sites in 2021. (c, f, i) Time series of HCHO and total observed VOCs with their correlation coefficient inserted.

Yes we checked and observed some spikes in VOC levels at the Tung Chung site on several days in Jan, which may be attributed to influences from some abrupt anthropogenic emissions. This was previously mentioned in Sect. 3.1 but has now been removed as suggested by reviewer #1. We ensured that these spikes are not artifacts of data collection process; rather, they are genuine measurement signals reflecting varying ambient conditions at this site and are thus retained for analysis.

Page19 Line8-15: Need to compare with more research on Hong Kong or the PRD region to better illustrate the sources of these underestimated species from models, or refer to more localized emission inventory data.
Response: The localized emission inventory developed by HKEPD has already been incorporated in our model simulations, as mentioned in P10 L10 that "The anthropogenic emissions used are the HKEPD emission inventory scaled to 2019 for HK region and scaled to 2021 for the PRD region …"

We have revised the discussion here and also added studies that focus on the PRD region. P20 L2: "Both ethane and propane are significantly affected by leakage from oil and natural gas production and use (**Ou et al., 2018; Liu et al., 2008;** Ge et al., 2024; Guo et al., 2007), resulting in a strong correlation between the two (Fig. S9). The model accurately captures such correlation; however, ethane (ETHA) concentrations and variability are better simulated than those of propane (Fig. S9). **This underestimation of propane, while ethane is reasonably simulated, has also been observed in various regions around the world (Ge et al., 2024; Rowlinson et al., 2024; Adedeji et al., 2023). Consequently, the model exhibits lower propane-to-ethane ratios compared to the measurements, suggesting the presence of unaccounted sources of propane emissions.**"

Page 22  Line 7-15: These sentences read a bit awkwardly. What is the FNR threshold used in this study to divide VOC-limited and transitional regime sensitive to both VOC and NOx, and which season in Figure 9 corresponds to different months? What is the threshold in other literature and how much does it differ from the PRD? These information need to be provided by the author in the text here, rather than simply expressing regional specificity. Meanwhile, it is difficult to obtain the specific values of FNR values for each month on the vertical axis of Figure 9c. It is recommended to modify it.

Furthermore, the FNR threshold used in this study is based on research in PRD region. It is necessary to verify whether it is fully applicable in different seasons and weather conditions, and to discuss the influence of the satellite inversion uncertainties of $NO_2$ and HCHO on the FNR classification. It is suggested to conduct an uncertainty propagation analysis of the FNR threshold to assess its impact on the ozone sensitivity classification. At the same time, it is suggested to compare the data from Hong Kong with that from other cities in the PRD region, in order to verify the regional consistency in the composition of VOCs and the ozone sensitivity.

Response:

(1) FNR thresholds used in this study are given in Figure 9 caption. We have now added this information in the main text in P22 L15, as in '**According to a set of region- and season-specific FNR thresholds derived for the PRD (spring and summer [1.55, 3.05], autumn [1.45, 2.95], and winter [1.65, 3.15]; Wang Y. et al., 2023a), we find that…**'.

Season definition has now been addressed in Figure 9 caption: "**Spring covers March–May, summer covers June–August, autumn covers September–November, and winter covers December–February.**"

Thresholds from other studies have now been added in P22 L20, as in 'other studies have also derived FNR thresholds but for China as a whole, **including [2.2, 3.2] from Ren et al. (2022), [2.3, 4.2] from Wang et al. (2021), and [1.0, 1.9] from Li et al. (2021). Our PRD region-specific thresholds from Wang Y. et al. (2023a) fall within the ranges established by the aforementioned independent studies, but for all of China.**'.

Specific values of FNR for each month are now added to Figure 9c.

(2) As above, this study adopted season-specific FNR thresholds, rather than uniform FNR thresholds across different seasons.

We have now analyzed how different satellite inversion products affect our analysis, as well as how uncertainty in different retrievals propagates to affect regime classification. P24 L35: "**We note that regimes can vary depending on the choice of satellite inversion products and FNR thresholds. To address this, we analyzed different spaceborne $NO_2$ products with relative differences ranging from –19% to 50% (25th–75th percentile) in the domain; yet these differences diminished in the regime classification, resulting 5% of grids showing discrepancies between the two spaceborne products (Fig. S11).** We also evaluated various FNR thresholds derived from Wang Y. et al. (2023a) and other independent studies (Fig. S12; Fig. S13). **All results consistently indicate that the model simulates a higher prevalence of VOC-limited regions compared to TROPOMI, despite the variations in satellite inversion products and FNR thresholds applied in regime classifications.**"

[Figure]

Figure S11. Differences between two TROPOMI NO2 products retrieved by (a) Product Algorithm Laboratory (PAL) and (b) Peking University (PKU; Liu et al., 2020), along with their associated FNR and ozone formation regimes.

Regional consistency has been examined, as the reviewer suggested. The two Hong Kong sites show a similar VOC composition to other sites across the PRD region (Fig. R2).

[Figure]

Figure R2. VOC composition at various sites across the PRD region during September 4–5, 2022.

Page 24 Line1-15: Many of the text here are meaningless. Please add some quantitative summary of the data, the existing research's understanding of underestimation, and the differences from this study. Also, summarize how future improvements in the model can optimize this situation. Additionally, the underestimation of OVOCs in the model has been proposed by many studies. The CMAQ model in this study is based on very few species, and underestimation is inevitable. It is suggested to add more detailed chemical mechanisms of OVOCs (such as MCM or updated aromatics oxidation mechanisms) and conduct comparative analysis.

Response:

(1) The main purpose here is to illustrate how the underrepresentation of VOCs and the resulting smaller FNR simulated in models, as revealed in the last section, can impact ozone management strategies. This aligns with our manuscript title regarding the role of VOCs in ozone pollution management. In response to the reviewer's suggestion, we have relocated less relevant content from this paragraph to other sections. Please refer to the new version in P25 L8.

(2) We have now added Table S1 to show quantitative comparisons between our observation-to-model differences with those from previous studies, and have addressed these comparisons in the main text.

P19 L11: "**Our observation-to-model differences are comparable to those reported in previous studies that primarily focus on NMHCs; however, our findings also emphasize a greater presence of oxygenated compounds that were not addressed in those studies (Table S1).**"

Table S1. Comparison between observed VOCs and modeled counterparts by 3-D chemical transport models. Negative values indicate model underestimation.

| | Ge et al. (2024) | Rowlinson et al. (2024) | | | | | | She et al. (2024) | Chen et al. (2019) | This study | |
|---|---|---|---|---|---|---|---|---|---|---|---|
| | Europe | Europe | North America | Southern Hemisphere | Pacific Ocean | Atlantic Ocean | Asia | China | USA | HK (land) | HK (water) |
| Unspeciated VOC total | | | | | | | | –30% | –37% | –47% for NMHCs; –45% for OVOCs | –48% for NMHCs; –70% for OVOCs |
| Ethyne (C2H2) | –7% ~ –13% | | | | | | | –45% | | –88% | –73% |
| Ethene (C2H4) | –5% ~ –29% | | | | | | | –40% | | –19% | –2% |
| Isoprene (C5H8) | | | | | | | | | | –18% | –21% |
| Alpha-Pinene (C10H16) | | | | | | | | | | | –7% |
| Ethane (C2H6) | –12% ~ –14% | –4% ~ –38% | –4% ~ –38% | –23% ~ –32% | –2% ~ –34% | –16% ~ –50% | –19% ~ +26% | | | –16% | +9% |
| Propane (C3H8) | –49% ~ –56% | –39% ~ –60% | –45% ~ –64% | –78% ~ –79% | –38% ~ –56% | –56% ~ –76% | –32% ~ –49% | | | –57% | –76% |
| n-butane (nC4H10) | +45% ~ +55% | | | | | | | | | | |
| i-butane (iC4H10) | –30% ~ –38% | +24% ~ +45% | –4% ~ +3% | –60% | –10% ~ +5% | –12% ~ –15% | +55% ~ +86% | –41% | | | |
| n-Pentane (nC5H12) | +26% ~ +44% | | | | | | | | | | |
| i-Pentane (iC5H12) | –53% ~ –59% | | | | | | | | | | |
| n-Hexane (nC6H14) | –8% ~ +9% | | | | | | | | | | |

| | | | | | | | | | | |
|---|---|---|---|---|---|---|---|---|---|---|
| Benzene | −12% ~ −17% | | | | | | | | −13% | −34% |
| Toluene | −33% ~ −39% | | | | | | −60% | | | |
| Xylene | −13% ~ +3% | | | | | | | | +93% | −23% |
| HCHO | | | | | | | +66% | | −23% | −42% |
| Acetaldehyde | | | | | | | | | −58% | −57% |
| Aldehydes with 3 or more carbons | | | | | | | | | −53% | −69% |
| Acetone | | | | | | | | | +26% | −65% |
| Glyoxal | | | | | | | | | | +30% |
| Methylglyoxal | | | | | | | | | | −27% |

(3) The contents on future model improvements are already presented in Sect. 6 (Conclusion). As each of Sections 3–5 represents observation-to-model comparisons from different sources (land-based in Sect. 3; shipborne in Sect. 4; spaceborne in Sect. 5), it is more logical to present future model development in Sect. 6 (Conclusion) as a summary.

Section 6 P26 L17: "Moving forward, there is a growing need to refine the speciation profiles (Rowlinson et al., 2024; Ge et al., 2024) and unaccounted sources (She et al., 2024; Travis et al., 2024; Beaudry et al., 2025) of VOC emissions, as well as to dissect a more detailed breakdown of the chemical mechanisms for oxygenated species (Travis et al., 2024; Gao et al., 2024) to improve model representation."

(4) We have a separate CMAQ simulation using the SAPRC07 mechanism, which was used here to compare with the CB6 mechanism used in our study. Preliminary results indicate an 11%–28% difference between the two schemes, as shown in Figure R3.

[Figure]

Figure R3. Comparison of hourly time series for SAPRC07 and CB06 during July 2021.

We have a separate study that specifically addresses the VOC differences simulated using SAPRC07 versus CB06. While this manuscript focuses on model comparisons with multi-platform observations, the details of intra-mechanism comparisons are beyond its scope and are therefore not included.

Specific Comments
Please check that the full names of many abbreviations do not require capitalization of the first letter. Pay attention to the format of the SI document, at least the title, author, and other information are required.
Response: Revised as suggested. Information has been added to the SI cover page in accordance with the ACP format.

Page 2 Line 4: would suggest remove "first time". This manuscript talked a lot about first time, if so, why wasn't previous research conducted? Is there any difficulty? All of these need to be raised. But in fact, this has already been done elsewhere.

Response: To our knowledge, this study marks the first shipborne measurements over Hong Kong waters, which enables the first integration of multi-source observations from shipborne, on-land, and spaceborne platforms to examine region air pollution in the PRD region. We have now clarified it as '**first attempt to integrate multi-source observations from on-land, shipborne, and spaceborne platforms with chemical transport modeling to examine VOC and ozone pollution in the PRD region of South China**', as specified in P4 L31. We removed the 'first time' expression from the abstract (P2 L4) due to space constraints.

If the reviewer could kindly provide references for any previous work that has integrated these three types of measurements in the PRD region, we would be happy to revise our statements accordingly.

Page2 Line 18: TROPOMI here should be modified as full name as TROPOspheric Monitoring Instrument (TROPOMI).

Response: Revised as suggested.

Page4 Line 21: Please check the references here. Wang et al., 2024, have 2 articles in the references section.

Response: Corrected as suggested.

Page 3 Line 7: The full name of HKEPD appears repeatedly, please review the entire manuscript for similar issues, such as the full name of the VCDs.

Response: We ensured that the full names of HKEPD and VCD are only provided the first time they appear and in standalone figure captions.

Page 3 Line 10: Fig S1 needs to add captions indicating the types of stations marked differently on the figure.

Response: Revised as suggested.

Page 4 Line24-31: I think the description here is pointless. Would suggest summarize and rewording these sentences. Some of the content should be in Section 2.

Response: We have now moved significant portion on introducing field measurements into Section 2, as suggested.

Introduction P4 L25: "**To address these gaps, this study integrated field observations, from a joint field campaign conducted by the Hong Kong Environmental Protection Department (HKEPD) and the Hong Kong University of Science and Technology (HKUST) during 2021–2022, with spaceborne TROPOMI data to…**"

Method P5 L3: "**The HKEPD–HKUST joint field campaign produced a comprehensive suite of surface measurements, including year-round continuous land-based monitoring (Mai et al., 2024; Lin et al., 2021) and the first shipborne mobile air quality observations over Hong Kong waters (Sun et al., 2024; Xu et al., 2023). A wide range of VOC species was measured, with 45 species (16 oxygenated) on land and 98 species (35 oxygenated) over water.**"

Page 4 line36: I think it is not advisable to use the results of this article to represent the entire southern China, would suggest modified to PRD region in China.

Response: Revised as suggested.

Page 6 Line 10-11: GC-MS/ECD/FID, need to provide the full name here.
Response: Full name is now provided, as suggested.

Page 8 Line 17: The full name of CMAQ is not reflected and the abbreviation of line 26 CMAQ should be before the Modeling System.
Response: We have now revised the expression to be 'The Community Multiscale Air Quality (CMAQ) modeling system'. This paragraph is now moved to Section 2.3, after the introduction of the CMAQ model.

Page 12: Is the NMHCs in Figure 1d missing the data for December? Even so, the horizontal axis on the graph should be 1-12 instead of not displaying 12.
Response: Revised as suggested.

Page 13 Line 6: HCHO-to-NO2 ratio (FNR), it needs to be clarified that F refers to formaldehyde.
Response: Revised as suggested.

Page 14: It is recommended to use the form of year/month or month number instead of the word 'month' for the horizontal axis in Figure 3 and Figure S4. Additionally, please do not use screenshot format in Figure (unnecessary lines appear in Figures S4 and S5).
Response: Revised as suggested.

Page 17: The wavy line in Figure 5 maybe an incorrect format.
Response: Revised as suggested.

---

## Author Response (AR2)

**Author Responses to Reviewer #2's Comments**

The authors have made many significant improvements to the manuscript, and addresses many of the comments in my previous review. I appreciate the authors works, and I am satisfied by the responses. Overall, I support publication. I have also made a number of comments on the new content that I think will help to clarify the material and Figure.

We would like to thank the reviewer for the thoughtful and insightful comments, which improved this manuscript. We have further revised the manuscript according to the reviewer comments.

General comments
Pg 6 line 32-Pg 7 line7: The use of different analytical methods for land-based and shipborne OVOC measurements introduces uncertainty and makes direct comparisons less robust. Please provide a more nuanced discussion of the limitations posed by different OVOC measurement techniques in manuscript.

Response: We did not directly compare land-based and shipborne measurements, as we are aware of such intra-instrumental differences. The goal of this paper is to reveal VOC representation in the model; therefore, we compare multi-platform observations individually with model rather than among the observations themselves. To address the limitations posed by different OVOC measurement techniques, we have added the statement, "**Different OVOC measurement techniques can introduce uncertainties (Cui et al., 2016; Wisthaler et al., 2008)**". We also referenced our peer reviewed publications which provides detailed comparisons between instruments: "**readers interested in detailed QA/QC methodologies are recommended to refer to the Supplement Text or the original publications listed above**".

Pg 25-Pg 28: I think the Section Conclusion is overly lengthy. Would suggest summarizing and condensing some of the text. For instance, comparisons or references to existing studies should not be presented here. Additionally, the manuscript clearly shows the model underestimates OVOCs but does not sufficiently discuss the specific reasons (emissions vs. chemistry), leaving a gap in the narrative. Please clarify the potential causes of model OVOC underestimation.

Response: Following reviewer suggestion, we have now trimmed the text by removing unnecessary content and references. Reasons are discussed in third paragraph of Conclusion Section.

Specific Comments
Pg1 line 3: Please check the full name of the "NMHCs".
Response: Revised as suggested.

Pg1 line 27: Please check "VOC" here, maybe it should be "VOCs".
Response: Revised as suggested.

Pg 6 line 22: Please check the full name of the "MSD/ESD/FID" have appeared in the previous text.
Response: Revised as suggested.

Pg 6 line 28: Please check the full name of the "UHPLC /UHPLC-MS" have appeared in the previous text.
Response: Revised as suggested.

Pg 9 Fig.1: Please add the east longitude and north latitude on the coordinate axis of the Fig.
Response: Revised as suggested.

Pg 12 Fig.2: "NO2 and O3" Please note that the numbers here should be subscripts.
Response: Revised as suggested.

Pg 14 Fig.3(c), (f), (i): Please add more subdivisions on the horizontal and vertical axes, where the horizontal axis should correspond to at least the data for each month. The resolution of the data in the graph is not monthly. It is recommended to replace the current presentation with a date or better format. And bold data lines on the graph would be better.
Response: Revised as suggested.

Pg 15 line 14: Please check OPF here should be OFP.
Response: Revised as suggested.

Pg 17 Fig.5: Please add more subdivisions on the horizontal and vertical axes.
Response: Revised as suggested.

Pg 18 Fig 6: Please add more subdivisions on the horizontal and vertical axes. "NO2 and O3" Please note that the numbers here should be subscripts.
Response: Revised as suggested.

Pg 18 Fig 7: It is difficult to understand just giving the abbreviations of species, and the difference in color between different species is not significant, and it is recommended to use the same color for same composition group VOC species. It is also suggested to rearrange each species in an order (composition or concentration).
Response: We have now added full names for each species and reorganized them by VOC sub-groups from alkanes, alkenes, alkynes, terpenes, aromatics, aldehydes, and ketones. We compared this color scheme with the option of color-coding by VOC sub-groups and found that the current scheme offers clearer visualization, so we have decided to retain it.

Pg 23 Fig.10: Please add the east longitude and north latitude on the coordinate axis of the Fig.
Response: Revised as suggested.

Pg 23 Line 3: Please check the full name of the "VCDs" have appeared in the previous text.
Response: Revised as suggested.

SI:
Pg 4: Fig.S3: The thickness of the horizontal axis lines on the graph is not uniform, and add more subdivisions on the horizontal axes will be better.
Response: Revised as suggested.

Pg 5 Fig.S4: Please add more subdivisions on the horizontal and vertical axes.
Response: Revised as suggested.

Pg 6 Fig.S5: Please add more subdivisions on the horizontal and vertical axes. "O3" Please note that the numbers here should be subscripts, and "VOC" should be VOCs.
Response: Revised as suggested.

Pg 6 Fig.S6: Please add the east longitude and north latitude on the coordinate axis of the Fig. Same as Fig.S11-13.
Response: Revised as suggested.

Pg 7 Fig.S7: Please add more subdivisions on the horizontal and vertical axes. The range of the vertical axis on the Fig. S7(a) should start from 0.
Response: Revised as suggested. We intentionally set the starting value of the y-axis to 20 in (a) for better visualization.